# Dynamical Low-Rank Compression of Neural Networks with Robustness under Adversarial Attacks

**Steffen Schotthöfer**,[*] **H. Lexie Yang**,[†] and **Stefan Schnake**[*]

[*]Computer Science and Mathematics Division, [†]Geospatial Science and Human Security Division
Oak Ridge National Laboratory
Oak Ridge, TN 37831 USA
`{schotthofers,yangh,schnakesr}@ornl.gov`

## Abstract

Deployment of neural networks on resource-constrained devices demands models that are both compact and robust to adversarial inputs. However, compression and adversarial robustness often conflict. In this work, we introduce a dynamical low-rank training scheme enhanced with a novel spectral regularizer that controls the condition number of the low-rank core in each layer. This approach mitigates the sensitivity of compressed models to adversarial perturbations without sacrificing accuracy on clean data. The method is model- and data-agnostic, computationally efficient, and supports rank adaptivity to automatically compress the network at hand. Extensive experiments across standard architectures, datasets, and adversarial attacks show the regularized networks can achieve over 94% compression while recovering or improving adversarial accuracy relative to uncompressed baselines.

## 1 Introduction

Deep neural networks have achieved state-of-the-art performance across a wide range of tasks in computer vision and data processing. However, their success comes at a cost of substantial computational and memory demands, which hinders deployment in resource-constrained environments. While significant progress has been made in scaling up models through data centers and specialized hardware, a complementary and equally important challenge lies in the opposite direction: deploying accurate and robust models on low-power platforms such as unmanned aerial vehicles (UAVs) or surveillance sensors. These platforms often operate in remote locations with limited power and compute resources, and are expected to function autonomously over extended periods without human intervention.

This setting introduces three interdependent challenges:

- **Compression:** Models must operate under strict memory, compute, and energy budgets.
- **Accuracy:** Despite being compressed, models must maintain high performance to support critical decision-making.
- **Robustness:** Inputs may be corrupted by noise or adversarial perturbations, requiring models to be resilient under distributional shifts.

Recent work has shown that these three objectives are inherently at odds. Compression via low-rank [38] or sparsity techniques [14] often leads to reduced accuracy. Techniques to improve adversarial robustness—such as data augmentation [24] or regularization-based defenses [54]—frequently degrade clean accuracy. Moreover, it has been observed that low-rank compressed networks can exhibit increased sensitivity to adversarial attacks [35]. Finally, many methods to increase adversarial robustness of the model impose additional computational burdens during training [43, 8] or inference [9, 15, 28], further complicating deployment on constrained hardware.

**Our Contribution.** We summarize our main contributions as follows:

39th Conference on Neural Information Processing Systems (NeurIPS 2025).

- **Low-rank compression framework.** We introduce a novel regularization and integration method to modify a class of low-rank training methods that yields low-rank compressed neural networks, achieving a more than $10\times$ reduction in both memory footprint and compute cost, while maintaining clean accuracy and adversarial robustness on par with full-rank baselines.
- **Theoretical guarantees.** We analyze the proposed regularizer and derive an explicit bound on the condition number $\kappa$ of each regularized layer. The bound gives further confidence that the regularizer improves adversarial performance.
- **Preservation of performance.** We prove analytically—and verify empirically—that our regularizer neither degrades training performance nor reduces clean validation accuracy across a variety of network architectures.
- **Extensive empirical validation.** We conduct comprehensive experiments on multiple architectures and datasets, demonstrating the effectiveness, robustness, and broad applicability of our method.

Beyond these core contributions, our approach is model- and data-agnostic, can be integrated seamlessly with existing adversarial defenses, e.g., adversarial training [13], and never requires assembling full-rank weight matrices—the last point guaranteeing a low memory footprint during training and inference. Moreover, by connecting to dynamical low-rank integration schemes and enabling convergence analysis via gradient flow, we offer new theoretical and algorithmic insights. Finally, the use of interpretable spectral metrics enhances the trustworthiness and analyzability of the compressed models.

## 2 Controlling the adversarial robustness of a neural network through the singular spectrum of its layers

We consider a neural network $f$ as a concatenation of $L$ layers $z^{\ell+1} = \sigma^\ell(W^\ell z^\ell)$ with matrix valued[1] parameters $W^\ell \in \mathbb{R}^{n \times n}$, layer input $z^\ell \in \mathbb{R}^{n \times b}$ and element-wise nonlinear activation $\sigma^\ell$. For simplicity of notation, we do not consider biases, but they are included for the numerical experiments in Section 6. The data $X$ constitutes the input to the first layer, i.e. $z^0 = X$. We assume that the layer activations $\sigma^\ell$ are Lipschitz continuous, which is the case for all popular activations [35]. The network is trained on a loss function $\mathcal{L}$ which we assume to be locally bounded with a Lipschitz continuous gradient. Throughout this work, we call a network in the standard format a "baseline" network.

**Low-rank Compression:** The compression the network for training and inference is typically facilitated by approximating the layer weight matrices by a low-rank factorization $W^\ell = U^\ell S^\ell V^{\ell,\top}$ with $U^\ell, V^\ell \in \mathbb{R}^{n \times r}$ and $S^\ell \in \mathbb{R}^{r \times r}$, where $r \leq n$ is the rank of the factorization. In this work, we generally assume that $U^\ell, V^\ell$ are orthonormal matrices at all times during training and inference. This assumption deviates from standard low-rank training approaches [17], however recent literature provides methods that are able to fulfill this assumption approximately [55] and even exactly [38, 37]. If $r \ll n$, the low-rank factorization with a storage and matrix-vector computation cost cost of $\mathcal{O}(2nr + r^2)$ is computationally more efficient than the standard matrix format $W$ with a computational cost of $\mathcal{O}(n^2)$.

**Adversarial robustness:** The adversarial robustness of a neural network $f$, a widely used trustworthiness metric, can be measured by its relative sensitivity $\mathcal{S}$ to small perturbations $\delta$, e.g., noise, of the input data $X$ [49, 11], i.e., $\mathcal{S}(f, X, \delta) := \frac{||f(X+\delta)-f(X)||}{||f(X)||} \frac{||X||}{||\delta||}$. In this work, we consider the sensitivity in the Euclidean ($\ell^2$) norm, i.e., $||\cdot|| = ||\cdot||_2$. For neural networks consisting of layers with Lipschitz continuous activation functions $\sigma^\ell$, $\mathcal{S}$ can be bounded [35] by the product

$$\mathcal{S}(f, X, \delta) \leq \big(\textstyle\prod_{\ell=1}^{L} \kappa(W^\ell)\big)\big(\textstyle\prod_{\ell=1}^{L} \kappa(\sigma^\ell)\big) \tag{1}$$

where $\kappa(W) := ||W|| \, ||W^\dagger||$ is the condition number of a matrix $W$, $W^\dagger$ is the pseudo-inverse of $W$, and $\kappa(\sigma)$ is the condition number of the layer activation function $\sigma$. The condition number of the element-wise non-linear activation functions $\sigma^\ell$ can be computed with the standard definitions (see [45] and [35] for condition numbers of several popular activation functions). Equation (1) allows us to consider each layer individually, thus we drop the superscript $\ell$ for brevity of exposition.

---

[1]We provide an extension to tensor-valued layers, e.g. in CNNs, in Section 5.1

[2]Note that the difference between the baseline and low-rank singular spectrum may be less pronounced for other layers and architectures. However, we have observed in all test cases that regularization with $\mathcal{R}$ makes the singular spectrum of the low-rank network more benign.

The sensitivity of a low-rank factorized network can be readily deducted from Equation (1) by leveraging orthonormality of $U$ and $V$, i.e., $\kappa(USV^\top) = \kappa(S)$. Thus, we only consider the $r \times r$ coefficient matrix $S$ to control the sensitivity of the network. The condition number $\kappa(S)$ can be determined via a singular value decomposition (SVD) of $S$, which is computationally feasible when $r \ll n$.

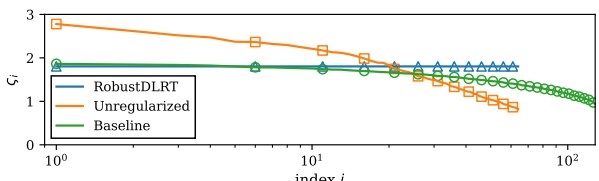

Figure 1: The singular values $\varsigma_i(W)$ of sequential layer 7 in VGG16 for baseline training, unregularized dynamical low-rank training, and RobustDLRT with our condition number regularizer $\mathcal{R}$ with $\beta = 0.075$ (see Section 5). The matrix $W$ is formed as the first-mode unfolding of the convolutional tensor. Conditioning of the regularized low-rank layer is significantly improved compared to the non-regularized low-rank and baseline layer.[2]

**Adversarial robustness-aware low-rank training:** Enhancing the adversarial robustness of the network during low-rank training thus boils down to controlling the conditioning of $S$, which is a non-trivial task. Moreover, the dynamics of the singular spectrum of $S$ of adaptive low-rank training schemes as Dynamical Low-Rank Training (DLRT) [38] become more ill-conditioned than the baseline during training, even if $S$ is always full rank. In Figure 1, we observe that the singular values $\varsigma$ of a rank 64 factorization of a network layer compressed with DLRT range from $\varsigma_{r=1} = 2.7785$ to $\varsigma_{r=64} = 0.8210$ yielding a condition number of $\kappa(S) = 3.3844$. In comparison, the baseline network has singular values ranging from $\varsigma_{r=1} = 1.8627$ to $\varsigma_{r=128} = 0.9445$ yielding a lower condition number of $\kappa(S) = 1.9722$. As a result, an $\ell^2$-FGSM attack with strength $\epsilon = 0.3$, reduces the accuracy of the baseline network to $54.96\%$, while the accuracy of the low-rank network drops to $43.39\%$, see Table 2.

# 3 Related work

**Low-rank compression** is a prominent approach for reducing the memory and computational cost of deep networks by constraining weights to lie in low-rank subspaces. Early methods used post-hoc matrix [12] and tensor decompositions [23], while more recent approaches enforce low-rank constraints during training for improved efficiency and generalization.

Dynamical Low-Rank Training [38] constrains network weights to evolve on a low-rank manifold throughout training, allowing substantial reductions in memory and FLOPs without requiring full-rank weight storage. The method has been extended to tensor-valued neural network layers [53], and federated learning [36]. Pufferfish [47] restricts parameter updates to random low-dimensional subspaces, while intrinsic dimension methods [2] argue that many tasks can be learned in such subspaces. GaLore [56] reduces memory cost by projecting gradients onto low-rank subspaces.

In contrast, low-rank fine-tuning methods like low-rank adaptation (LoRA) [17] inject trainable low-rank updates into a frozen pre-trained model, enabling efficient adaptation with few parameters. Extensions such as GeoLoRA [37], AdaLoRA [55], DyLoRA [46], and DoRA [31] incorporate rank adaptation or structured updates, improving performance over static rank baselines. However, these fine-tuning methods do not reduce the cost of the full training and inference, thus are not applicable to address the need of promoting computational efficiency.

**Pruning** is another well studied approach to reduce the number of parameters of a trained neural network [18, 26, 40, 57, 7, 19] by either sparsifying weight matrices or layer output channels of a network. Typically sparsity pruning is performed after training a fully parametrized neural network and thus only reduces memory and compute load during inference, while treating training as an offline cost.

**Improving adversarial robustness** with orthogonal layers has been a recently studied topic in the literature [3, 4, 48, 10, 35]. Many of these methods can be classified as either a soft approach, where orthogonality is imposed weakly via a regularizer, or a hard approach, where orthogonality is explicitly enforced in training.

Examples of soft approaches include the soft orthogonal (SO) regularizer [48], double soft orthogonal regularizer [4], mutual coherence regularizer [4], and spectral normalization [32]. These regularization-based approaches have several advantages; namely, they are more flexible to many

problems/architectures and are amenable to transfer learning scenarios (since pertained models are admissible in the optimization space). However, influencing the spectrum weakly via regularization cannot enforce rigorous and explicit bounds on the spectrum.

Many hard approaches strongly enforce orthogonality/well-conditioned constraints by training on a chosen manifold using Riemannian optimization methods [25, 1, 35]. A hard approach built for low-rank training is given in [35]; this method clamps the extremes of the spectrum to improve the condition number during training. The clamping gives a hard estimate on the range of the spectrum which enables a direct integration of the low-rank equations of motion with reasonable learning rates. However, this method requires a careful selection of the rank $r$, which is viewed as a hyperparameter in [35]. If $r$ is chosen incorrectly, the clamping of the spectrum, a hard-thresholding technique, acts as a strong regularizer which could affect the validation metrics of the network.

Our regularization method detailed below falls neatly into a soft approach and our proposed regularizer can be seen as an extension of the soft orthogonality (SO) regularizer [48] to well-conditioned matrices in the low-rank setting. As noted in [4], the SO regularizer only works well when the input matrix is of size $m \times n$ with $m \leq n$. However, we avoid this issue since the regularizer is applied to the square $r \times r$ matrix $S$; an extension to convolutional layers is discussed in Section 5.1. In the context of low-rank training, the soft approach enables rank-adaptivity of the method.

## 4   Improving conditioning via regularization

We design a computationally efficient regularizer $\mathcal{R}$ to control and decrease the condition number of each network layer during training. The regularizer $\mathcal{R}$ only acts on the small $r \times r$ coefficient matrices $S$ of each layer and thus has a minimal memory and compute overhead over low-rank training. The regularizer is differentiable almost everywhere and compatible with automatic differentiation tools. Additionally, $\mathcal{R}$ has a closed form derivative that enables an efficient and scalable implementation of $\nabla\mathcal{R}$. Furthermore, $\mathcal{R}$ is compatible with any rank-adaptive low-rank training scheme that ensures orthogonality of $U, V$, e.g., [55, 36, 37, 35].

**Definition 1.** *We define the robustness regularizer $\mathcal{R}$ for any $S \in \mathbb{R}^{r \times r}$ by*

$$\mathcal{R}(S) = \|S^\top S - \alpha_S^2 I\|, \qquad where \qquad \alpha_S^2 = \frac{1}{r}\|S\|^2 \tag{2}$$

*and $I = I_r$ is the $r \times r$ identity matrix.*

The regularizer $\mathcal{R}$ can be viewed as an extension of the soft orthogonal regularizer [48, 4] where we penalize the distance of $S^\top S$ to the well-conditioned matrix $\alpha_S^2 I$. Here $\alpha_S$ is chosen such that $\|S\| = \|\alpha_S I\|$. Moreover, $\mathcal{R}$ is also a scaled standard deviation of the squared singular values $\{\varsigma_i(S)^2\}_{i=1}^r$, i.e.,

$$\frac{1}{r}\mathcal{R}(S)^2 = \frac{1}{r}\sum_{i=1}^r (\varsigma_i(S)^2)^2 - \left(\frac{1}{r}\sum_{i=1}^r \varsigma_i(S)^2\right)^2. \tag{3}$$

See Appendix C for the proof. Therefore, $\mathcal{R}$ is a unitarily invariant regularizer; namely, $\mathcal{R}(USV^\top) = \mathcal{R}(S)$ for orthogonal $U, V$. These two forms of $\mathcal{R}$ are useful in the properties shown below.

**Proposition 1.** *The gradient of $\mathcal{R}$ in (2) is given by $\nabla\mathcal{R}(S) = 2S(S^\top S - \alpha_S^2 I)/\mathcal{R}(S)$.*

See Appendix C for the proof. The gradient computation consists only of $r \times r$ matrix multiplications and a Frobenius norm evaluation. Thus $\nabla\mathcal{R}$ is computationally efficient for $r \ll m$. Further, its closed form enables a straight-forward integration into existing optimizers such as Adam or SGD applied to $S$.

**Proposition 2** (Condition number bound). *For any $S \in \mathbb{R}^{r \times r}$ there holds*

$$\kappa(S) \leq \exp\left(\frac{1}{\sqrt{2}\varsigma_r(S)^2}\mathcal{R}(S)\right). \tag{4}$$

See Appendix C for the proof. Thus, if $\varsigma_r(S)$ is not too small, we can use $\mathcal{R}(S)$ as a good measure for the conditioning of $S$. Note that the

Table 1: VGG16 on UCM data. Comparison of regularized LoRA and DLRT trained networks under the $\ell^2$-FGSM attack. Orthogonality of $U, V$ increases adversarial performance significantly.

| Method | c.r. [%] | clean Acc [%] | $\ell^2$-FGSM, $\epsilon = 0.1$ |
|---|---|---|---|
| Non-regularized DLRT | 95.30 | 93.92 | 72.41 |
| RobustDLRT, $\beta = 0.075$ | 95.84 | 94.61 | 78.68 |
| LoRA, $\beta = 0.075$ | 95.83 | 88.57 | 73.81 |

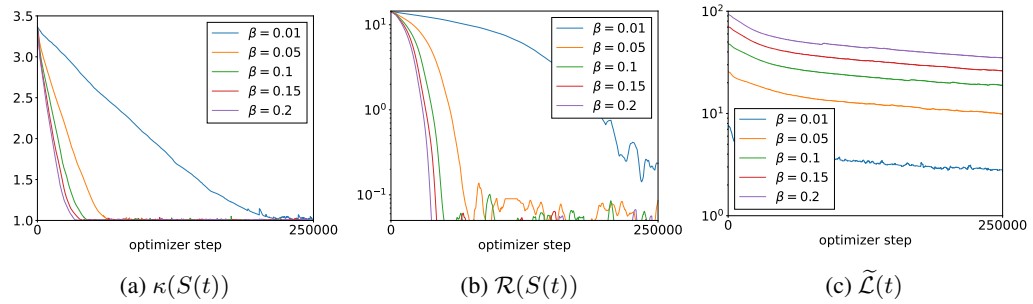

(a) $\kappa(S(t))$        (b) $\mathcal{R}(S(t))$        (c) $\widetilde{\mathcal{L}}(t)$

Figure 2: UCM Dataset, $\kappa(S(t))$ and $\mathcal{R}(S(t))$ of layer 4 of VGG16 for different regularizations strengths $\beta$. Each line is the median of 5 training runs. Higher $\beta$ values lead to faster reduction of the layer condition $\kappa(S)$, which quickly approaches its minimum value 1, and faster decay of $\mathcal{R}$. Unregularized training ($\beta = 0$) leads to $\kappa(S) > 1000$ after a few iterations.

singular value truncation used in rank-adaptive methods ensures that $\varsigma_r(S)$ is always sufficiently large. Figures 2a and 2b show the dynamics of $\mathcal{R}(S(t))$ and $\kappa(S(t))$ during low-rank regularized training; we see that $\kappa(S(t))$ decays as $\mathcal{R}(S(t))$ decays, validating Proposition 2.

**Remark 1.** *When $U, V$ are not orthonormal, e.g., in simultaneous gradient descent training (LoRA), the smallest $n - r$ singular values of $USV^\top$ are often zero-valued; thus, the bound of Equation* (4) *is not useful. Table 1 shows that the clean accuracy and adversarial accuracy of regularized LoRA is significantly lower than standard or regularized training with orthonormal $U, V$.*

We now study the stability of the regularizer when applied to a least squares regression problem, i.e., given a fixed $M \in \mathbb{R}^{r \times r}$ we seek to minimize $\mathcal{J}(S) := \beta \mathcal{R}(S) + \frac{1}{2}\|S - M\|^2$ over $S \in \mathbb{R}^{r \times r}$.

**Proposition 3.** *Consider the dynamical system generated by the gradient flow of $\mathcal{J}$; namely, $\dot{S}(t) + \beta \nabla \mathcal{R}(S(t)) + S(t) = M$. Then for any $t \geq 0$ we have the long-time stability estimate*

$$\tfrac{1}{2}\|S(t) - M\|^2 + 2\beta \int_0^t e^{\tau - t} \mathcal{R}(S(\tau))\, \mathrm{d}\tau \leq \tfrac{1}{2}e^{-t}\|S(0) - M\|^2 + 2(1 - e^{-t})\beta(1 + 2\beta)\|M\|^2. \quad (5)$$

See Appendix C for the proof. We note that unlike standard ridge and lasso regularizations methods, $\mathcal{R}$ lacks convexity; thus long-time stability of the regularized dynamics is not obvious. However, $\nabla \mathcal{R}$ possesses monotonicity properties that we leverage to show in (5) that the growth in $\mathcal{J}$ only depends on $\beta$, $M$, and the initial loss. Moreover, for large $t$, the change in the final loss by the regularizer only depends on $\beta$ and the true solution $M$ and not the specific path $S(t)$. While training on the non-convex loss will not provide the same theoretical properties as the convex least-square loss used in Proposition 3, the experiments in Figure 2 give confidence that adding our regularizer does not yield a relatively large change in the loss decay rate over moderate training regimes. Particularly, we observe empirically in Figure 2 that the condition number $\kappa(S)$ of decreases alongside the regularizer value $\mathcal{R}$ during training.

**Remark 2.** *We note $\mathcal{R}^2$ can also be used in place of $\mathcal{R}$. While $\mathcal{R}^2$ is differentiable at $\mathcal{R}(S) = 0$, we choose $\mathcal{R}$ as our regularizer due to the proper scaling in* (4).

## 5    A rank-adaptive and adversarial robustness increasing dynamical low-rank training scheme

In this section we integrate the regularizer $\mathcal{R}$ into a rank-adaptive, orthogonality preserving, and efficient low-rank training scheme. We are specifically interested in a training method that 1) enables separation of the spectral dynamics of the coefficients $S$ from the bases $U, V$ and 2) ensures orthogonality of $U, V$ at all times during training to obtain control layer conditioning in a compute and memory efficient manner. Popular schemes based upon simultaneous gradient descent of the low-rank factors such as LoRA [17] are not suitable here. These methods typically do not ensure orthogonality of $U$ and $V$. Consequently, $\mathcal{R}(USV^\top) \neq \mathcal{R}(S)$, and this fact renders evaluation of the regularizer $\mathcal{R}$ computationally inefficient.

Thus we adapt the two-step scheme of [36] which ensures orthogonality of $U, V$. The method dynamically reduces or increases the rank of the factorized layers depending on the training dynamics and the complexity of the learning problem at hand. Consequently, the rank of each layer is no longer a hyper-parameter that needs fine-tuning, c.f. [17, 35], but is rather an interpretable measure for the inherent complexity required for each layer.

To facilitate the discussion, we define $\widetilde{\mathcal{L}} = \mathcal{L} + \beta\mathcal{R}$ as the regularized loss function of the training process with regularization parameter $\beta > 0$. To construct the method we consider the (stochastic) gradient descent-based update of a single weight matrix $W_{t+1} = W_{t+1} - \lambda\nabla_W\widetilde{\mathcal{L}}$ for minimizing $\widetilde{\mathcal{L}}$ with step size $\lambda > 0$. The corresponding continuous time gradient flow reads $\dot{W}(t) = -\nabla_W\widetilde{\mathcal{L}}(W(t))$, which is a high-dimensional dynamical system with a steady state solution. We draw from established dynamical low-rank approximation (DLRA) methods, which were initially proposed for matrix-valued dynamical systems [20]. DLRA was recently extended to neural network training [38, 53, 36, 37, 22, 16] to formulate a consistent gradient flow evolution for the low-rank factors $U$, $S$, and $V$.

The DLRA method constrains the trajectory of $W$ to the manifold $\mathcal{M}_r$, consisting of $n \times n$ matrices with rank $r$, by projecting the full dynamics $\dot{W}$ onto the local tangent space of $\mathcal{M}_r$ via an orthogonal projection, see Figure 3. The low-rank matrix is represented as $USV^\top \in \mathcal{M}_r$, where $U \in \mathbb{R}^{n \times r}$ and $V \in \mathbb{R}^{n \times r}$ have orthonormal columns and $S \in \mathbb{R}^{r \times r}$ is full-rank (but not necessarily diagonal). An explicit representation of the tangent space leads to equations for the factors $U$, $S$, and $V$ in [20, Proposition 2.1]. However, following these equations requires a prohibitively small learning rate due to the curvature of the manifold [29]. Therefore, specialized integrators have been developed to accurately navigate the manifold with reasonable learning rates [29, 6, 5].

Below we list the method of [36] with the changes introduced by adding our robustness regularizer. We call the resulting scheme *RobustDLRT*, and a single iteration of RobustDLRT is specified in Algorithm 1.

**Basis Augmentation:** The method first augments the current bases $U^t, V^t$ at optimization step $t$ by their gradient dynamics $\nabla_U\mathcal{L}$, $\nabla_V\mathcal{L}$ via

$$
\begin{aligned}
\widehat{U} &= \texttt{orth}([U^t \mid \nabla_U\mathcal{L}(U^tS^tV^{t,\top})]) \in \mathbb{R}^{n \times 2r}, \\
\widehat{V} &= \texttt{orth}([V^t \mid \nabla_V\mathcal{L}(U^tS^tV^{t,\top})]) \in \mathbb{R}^{n \times 2r},
\end{aligned}
\tag{6}
$$

to double the rank of the low-rank representation and subsequently creates orthonormal bases $\widehat{U}, \widehat{V}$. Here $\texttt{orth}(A)$ denotes an orthonormal basis for the range of $A$ and $\mid$ denotes horizontal concatenation of matrices. Since $\mathcal{R}(USV^\top) = \mathcal{R}(S)$, $\nabla_U\mathcal{R}(USV^\top) = \nabla_V\mathcal{R}(USV^\top) = 0$; hence $\nabla_U\widetilde{\mathcal{L}} = \nabla_U\mathcal{L}$ and $\nabla_V\widetilde{\mathcal{L}} = \nabla_V\mathcal{L}$ are used in (6). The span of $\widehat{U}$ contains $U^t$, which is needed to ensure of the loss does not increase during augmentation, and a first-order approximation of $\text{span}(U^{t+1})$ using the exact gradient flow for $U$, see [36, Theorem 2] for details. Geometrically, the latent space

$$
\mathcal{S} = \{\widehat{U}Z\widehat{V}^\top : Z \in \mathbb{R}^{2r \times 2r}\}
\tag{7}
$$

can be seen as subspace[3] of the tangent plane of $\mathcal{M}_r$ at $U^tS^tV^{t,\top}$, see Figure 3.

**Latent Space Training:** We update the latent coefficients $\widehat{S}$ via a Galerkin projection of the training dynamics onto the latent space $\mathcal{S}$. The latent coefficients $\widehat{S}$ are updated by integrating the projected gradient flow

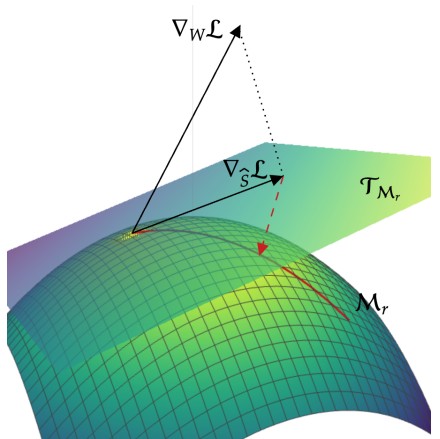

Figure 3: Geometric interpretation of Algorithm 1. First, we compute the parametrization of the tangent plane $\mathcal{T}_{\mathcal{M}_r}$. Then we compute the projected gradient update with $\nabla_{\widehat{S}}\mathcal{L}$. Lastly, we retract the updated coefficients back onto the manifold $\mathcal{M}_r$. The regularizer $\mathcal{R}$ steers training to regions of $\mathcal{M}_r$ with lower curvature.

---

[3]Technically the latent space contains extra elements not in the tangent space, but the extra information only helps the approximation.

---

**Algorithm 1:** Single iteration of RobustDLRT.

---

**Input :** Initial orthonormal bases $U, V \in \mathbb{R}^{n \times r}$ and diagonal $S \in \mathbb{R}^{r \times r}$;
$\vartheta$: singular value threshold for rank truncation; $\lambda$: learning rate.

1    Evaluate $\mathcal{L}(USV^\top)$                                   `/* Forward evaluate */`

2    $G_U \leftarrow \nabla_U \mathcal{L}(USV^\top); \ G_V \leftarrow \nabla_V \mathcal{L}(USV^\top)$         `/* Backprop on basis */`

3    $\widehat{U} \leftarrow \text{orth}([U \mid G_U]); \ \widehat{V} \leftarrow \text{orth}([V \mid G_V])$      `/* augmentation in parallel */`

4    $\widehat{S} \leftarrow \widehat{U}^\top U S V^\top \widehat{V}$                             `/* coefficient augmentation */`

5    $\widehat{S} \leftarrow \text{coefficient\_update}(\widehat{S}, s_*, \lambda, \beta)$    `/* regularized coefficient training */`

6    $U, S, V \leftarrow \text{truncation}(\widehat{S}, \widehat{U}, \widehat{V})$

7    **def** `coefficient_update(`$\widehat{S}_0$*: coefficient, $s_*$: # local steps, $\lambda$: learning rate, $\beta$: robustness*
       *regularization weight*)**:**

8       **for** $s = 1, \dots, s_*$ **do**

9          $G_S \leftarrow -\lambda \nabla_{\widehat{S}} \mathcal{L}(\widehat{U}\widehat{S}_{s-1}\widehat{V}^\top) - \beta \nabla_{\widehat{S}_s} \mathcal{R}(\widehat{S}_s)$

10        $\widehat{S}_s \leftarrow \widehat{S}_{s-1} + \text{optim}(G_S)$        `/* optimizer update, e.g., SGD or Adam */`

11      **return** $\widehat{S}_{s_*}$

12   **def** `truncation(`$\widehat{S}$*: augmented coefficient, $\widehat{U}$: augmented basis, $\widehat{V}$: augmented co-basis* )**:**

13      $P_{r_1}, \Sigma_{r_1}, Q_{r_1} \leftarrow$ truncated $\text{svd}(\widetilde{S})$ with threshold $\vartheta$ to new rank $r_1$

14      $U \leftarrow \widehat{U} P_{r_1}; \ V \leftarrow \widehat{V} Q_{r_1}$                        `/* Basis update */`

15      $S \leftarrow \Sigma_{r_1}$                  `/* Coefficient update with diagonal `$\Sigma_{r_1}$` */`

16      **return** $U, S, V$

---

$\dot{\widehat{S}} = -\widehat{U}^\top \nabla_W \widetilde{\mathcal{L}} \widetilde{\widehat{V}} = -\nabla_{\widehat{S}} \widetilde{\mathcal{L}}$ using stochastic gradient
descent or an other suitable optimizer for a number of $s_*$ local iterations, i.e.,

$$\widehat{S}_{s+1} = \widehat{S}_s - \lambda \nabla_{\widehat{S}} \mathcal{L} - \beta \nabla_{\widehat{S}} \mathcal{R}(\widehat{S}_s), \quad s = 0, \dots, s_* - 1. \tag{8}$$

Equation (8) is initialized with $\widehat{S}_0 = \widehat{U}^\top U^t S^t V^{t,\top} \widehat{V} \in \mathbb{R}^{2r \times 2r}$, and we set $\widetilde{S} = \widehat{S}_{s_*}$

**Truncation:** Finally, the latent solution $\widehat{U}\widetilde{S}\widehat{V}^\top$ is retracted back onto the manifold $\mathcal{M}_r$. The retraction can be computed efficiently by using a truncated SVD of $\widetilde{S}$ that discards the smallest $r$ singular values. To enable rank adaptivity, the new rank $r_1$ instead of $r$ can be chosen by a variety of criteria, e.g., a singular value threshold $\|[\varsigma_{r_1}, \dots, \varsigma_{2r}]\|_2 < \vartheta$. Once a suitable rank is determined, the bases $U$ and $V$ are updated by discarding the basis vectors corresponding to the truncated singular values.

**Remark 3.** *We note that $\mathcal{R}$ will likely increase the smallest singular values of $\hat{S}$ to improve $\kappa(\hat{S})$. This could theoretically increase the truncated rank over non-regularized DLRT and result in less compression. However, we find in the experiments in Section 6 that RobustDLRT has similar compression rates to DLRT.*

**Computational cost:** The computational cost of RobustDLRT is asymptotically the same as LoRA, since the reconstruction of the full weight matrix $W$ is never required. The orthonormalization, computation of the regularizer $\mathcal{R}$, and the SVD for accounts for $\mathcal{O}(nr^2)$, $\mathcal{O}(r^3)$, $\mathcal{O}(r^3)$ floating point operations, respectively. When using multiple coefficient update steps $s_* > 1$, the amortized cost is lower than that of LoRA, since only the gradient with respect to $\widehat{S}$ is required in most updates. While the regularizer may be applied to full-rank baseline models, its $\mathcal{O}(n^3)$ computational scaling significantly increases training costs.

### 5.1 Extension to convolutional neural networks

The convolution layer map in 2D CNNs translates a $W \times H$ image with $N_I$ in-features to $N_O$ out-features. Using tensors, this map is expressed as $Y = C * X$ where $X \in \mathbb{R}^{N_I \times W \times H}$, $Y \in \mathbb{R}^{N_O \times W \times H}$, and $C \in \mathbb{R}^{N_O \times N_I \times S_W \times S_H}$ is the convolutional kernel with a convolution window size $S_W \times S_H$. Neglecting the treatment of strides and padding, $C * X$ is given as a tensor contraction by

$$Y(o, w, h) = \sum_{c, s_w, s_h} C(o, c, s_w, s_h) X(c, w + s_w, h + s_h) \tag{9}$$

Table 2: UCM and Cifar10 benchmark. Clean and adversarial accuracy means and std. devs. of the baseline and regularized low-rank networks for different architectures. We report the low-rank results for $\beta = 0.0$ (DLRT) and the best performing $\beta$ that is given in Table 9. Algorithm 1 (RobustDLRT) is able to match or surpass baseline adversarial accuracy values at compression rates of up to $94\%$ in most setups. All runs where RobustDLRT surpasses the uncompressed baseline are highlighted.

| UCM Data | | | Clean | Acc [%] for $\ell^2$-**FGSM**, $\epsilon$ | | | Acc [%] for **Jitter**, $\epsilon$ | | Acc [%] for **Mixup**, $\epsilon$ | | |
| Method | c.r. [%] | | Acc. [%] | 0.05 | 0.1 | 0.3 | 0.035 | 0.045 | 0.025 | 0.1 | 0.75 |
|---|---|---|---|---|---|---|---|---|---|---|---|
| **VGG16** Baseline | 0.0 | | 94.40±0.72 | 86.71±1.90 | 76.40±2.84 | 54.96±2.99 | 89.58±2.99 | 85.05±3.40 | 77.77±1.61 | 37.25±3.66 | 23.05±3.01 |
| DLRT | 95.30 | | 93.92±0.23 | 87.95±1.02 | 72.41±2.08 | 43.39±4.88 | 83.99±1.22 | 67.41±1.63 | 85.79±1.51 | 40.42±2.89 | 20.13±2.92 |
| RobustDLRT | 95.84 | | 94.61±0.35 | 89.12±1.33 | 78.68±2.30 | 53.30±3.14 | 88.33±1.20 | 79.81±0.93 | 90.33±0.90 | 70.12±3.08 | 47.31±2.78 |
| **VGG11** Baseline | 0.0 | | 94.23±0.71 | 89.93±1.33 | 78.66±2.46 | 39.45±2.98 | 90.25±1.66 | 85.24±1.90 | 83.10±1.47 | 40.34±4.88 | 22.01±3.21 |
| DLRT | 94.89 | | 93.70±0.71 | 86.58±1.22 | 67.55±2.16 | 28.92±2.65 | 83.90±1.36 | 63.41±1.39 | 87.15±1.18 | 40.17±4.96 | 14.18±3.78 |
| RobustDLRT | 94.59 | | 93.57±0.84 | 87.90±0.91 | 72.96±1.55 | 32.85±2.46 | 86.77±0.76 | 74.31±1.50 | 88.00±1.13 | 60.97±4.18 | 28.56±3.64 |
| **ViT-16b** Baseline | 0.0 | | 96.72±0.36 | 93.02±0.38 | 92.18±0.31 | 89.71±0.28 | 93.71±1.22 | 93.21±1.17 | 89.62±1.81 | 51.05±3.17 | 43.91±3.97 |
| DLRT | 86.7 | | 96.38±0.60 | 91.21±0.44 | 82.10±0.32 | 62.45±0.41 | 86.67±1.05 | 79.81±0.81 | 80.48±1.82 | 41.52±3.24 | 35.91±3.76 |
| RobustDLRT | 87.9 | | 96.41±0.67 | 92.57±0.34 | 85.67±0.41 | 69.94±0.42 | 91.03±0.86 | 84.19±1.39 | 87.33±1.81 | 46.39±2.75 | 40.76±3.88 |
| Cifar10 Data | | | | | | | | | | | |
| **VGG16** Baseline | 0.0 | | 89.82±0.45 | 76.22±1.38 | 63.78±2.01 | 34.97±2.54 | 78.60±1.12 | 73.54±1.55 | 71.51±1.31 | 37.36±2.60 | 16.12±2.12 |
| DLRT | 94.37 | | 89.23±0.62 | 74.07±1.23 | 59.55±1.79 | 28.74±2.21 | 72.51±1.04 | 66.21±1.41 | 79.56±1.15 | 59.88±2.26 | 38.98±1.94 |
| RobustDLRT | 94.18 | | 89.49±0.58 | 76.04±1.18 | 62.08±1.69 | 32.77±2.04 | 75.53±0.98 | 69.93±1.22 | 87.62±1.07 | 84.80±2.01 | 81.26±2.15 |
| **VGG11** Baseline | 0.0 | | 88.34±0.49 | 75.89±1.42 | 64.21±1.96 | 31.76±2.45 | 74.96±1.09 | 68.59±1.63 | 74.77±1.26 | 40.88±2.58 | 08.95±1.98 |
| DLRT | 95.13 | | 88.13±0.56 | 72.02±1.34 | 55.83±1.92 | 21.59±2.16 | 66.98±1.05 | 58.57±1.55 | 79.42±1.08 | 47.95±2.18 | 22.92±1.77 |
| RobustDLRT | 94.67 | | 87.97±0.52 | 76.04±1.26 | 63.82±1.83 | 30.77±2.30 | 71.06±1.00 | 65.63±1.38 | 84.93±1.10 | 78.35±1.89 | 65.93±2.04 |
| **ViT-16b** Baseline | 0.0 | | 95.42±0.35 | 79.94±0.95 | 63.66±1.62 | 32.09±2.05 | 84.65±0.88 | 77.20±1.04 | 52.17±1.49 | 16.03±2.34 | 13.29±2.01 |
| DLRT | 73.42 | | 95.39±0.41 | 79.50±0.91 | 61.62±1.48 | 30.32±1.94 | 83.33±0.80 | 76.16±0.95 | 58.32±1.44 | 17.43±2.28 | 14.49±1.92 |
| RobustDLRT | 75.21 | | 94.66±0.38 | 82.03±0.88 | 69.29±1.43 | 38.05±1.99 | 87.97±0.75 | 83.03±0.91 | 74.49±1.32 | 27.80±2.11 | 18.34±1.87 |

where $s_w$ and $s_h$ range from $-S_W/2, \dots, S_W/2$ and $-S_H/2, \dots, S_H/2$ respectively, and $o = 1, \dots, N_O$, $w = 1, \dots, W$, and $h = 1, \dots, H$.

DLRT was extended to convolutional layers in [53] by compressing $C$ with a Tucker factorization. Little is gained in compressing the window modes as they are typically small. Thus, we only factorize $C$ in the feature modes with output and input feature ranks $r_O \ll N_O$ and $r_I \ll N_I$ as

$$C(o, i, s_w, s_h) = \sum_{q_O, q_I=1}^{r_I, r_O} U_O(o, q_O) U_I(i, q_I) S(q_O, q_I, s_w, s_h). \tag{10}$$

Substituting (10) into (9) and rearranging indices yields

$$Y(o, w, h) = \sum_{q_O} U_O(o, q_O) \widetilde{Y}(q_O, w, h), \tag{11a}$$

$$\widetilde{Y}(q_O, w, h) = \sum_{q_I, s_w, s_h} S(q_O, q_I, s_w, s_h) \widetilde{X}(q_I, w + s_w, h + s_h), \tag{11b}$$

$$\widetilde{X}(q_I, w + s_w, h + s_h) = \sum_c U_I(c, q_I) X(c, w + s_w, h + s_h). \tag{11c}$$

**Remark 4.** *Aside from the prolongation* (11a) *and retraction* (11c) *from/to the low-rank latent space, the low-rank convolution map* (11) *features a convolution* (11b) *similar to* (9) *but in the reduced dimension low-rank latent space.*

**Robustness regularization for convolutional layers.** The contractions in (9) and (11b) show that the output channels arise from a tensor contraction of the input channel and window modes; hence, both (9) and (11b) can be viewed as matrix-vector multiplications where $C$ is matricised on the output channel mode; i.e., $C \to \text{Mat}(C) \in \mathbb{R}^{N_O \times N_I S_W S_H}$ and $S \to \text{Mat}(S) \in \mathbb{R}^{r_O \times r_I S_W S_H}$. Therefore, we only regularize $\text{Mat}(S)$ with our robustness regularizer. Moreover, we assume $r_O \leq r_I S_W S_H$, which is almost always the case since $r_O$ and $r_I$ are comparable and $S_W S_H \gg 1$. Then we regularize convolutional layers by $\mathcal{R}(\text{Mat}(S)^\top)$ so that $SS^\top$ is an $r_O \times r_O$ matrix, which is computationally efficient.

We remark that the extension of Algorithm 1 to a tensor-valued layer with Tucker factorization only requires to change the truncation step; the SVD is replaced by a truncated Tucker decomposition of $S$. The Tucker bases $U_O$ and $U_I$ can be augmented in parallel similarly to the matrix case.

# 6 Numerical Results

We evaluate the numerical performance of Algorithm 1 compared with non-regularized low-rank training, baseline training, and several other robustness-enhancing methods the VGG16, VGG11, and ViT-16b architectures and University of California, Merced (UCM), Cifar10, and ImageNet1k datasets. Detailed descriptions of the models, datasets, pre-processing, training hyperparameters,

Table 3: Imagenet Benchmark, ViT-32l trained with baseline Adam, DLRT, and RobustDLRT. We report the low-rank results for unregularized $\beta = 0.0$ and the best performing $\beta$, given in Table 9. Algorithm 1 (RobustDLRT) is able to match or surpass baseline adversarial accuracy values in most setups. All runs where RobustDLRT surpasses the uncompressed baseline are highlighted.

| Method | c.r. [%] | Top1/Top5 Clean Acc. [%] | Top1/Top5 Acc [%] for $\ell^2$-**FGSM**, $\epsilon$ | | | Top1/Top5 Acc [%] for **Jitter**, $\epsilon$ | |
|---|---|---|---|---|---|---|---|
| | | | 0.05 | 0.1 | 0.3 | 0.035 | 0.045 |
| Baseline | 0 | 74.37/92.20 | 43.58/73.75 | 31.42/63.42 | 16.03/43.41 | 43.09/78.24 | 35.57/74.96 |
| DLRT | 58.02 | 72.27/90.06 | 42.70/70.43 | 30.32/60.90 | 15.47/40.58 | 43.98/74.49 | 38.44/ 71.31 |
| RobustDLRT | 57.98 | 72.25/90.03 | 43.17/71.58 | 35.11 /62.82 | 25.24/50.65 | 48.22 /77.35 | 43.51/75.14 |

and competitor methods are given in Appendix B. A reference implementation is provided at `https://github.com/ScSteffen/RobustDLRT`. We measure the compression rate (c.r.) as the relative amount of pruned parameters of the target network, i.e. c.r. $= (1 - \frac{\text{\#params low-rank net}}{\text{\#params baseline net}}) \times 100$. The reported numbers in the tables represent the average over 10 stochastic training runs. We observe in Table 2 that clean accuracy results exhibit a standard deviation of less than $0.8\%$; the standard deviation increases with the attack strength $\epsilon$ for all tests and methods. This observation holds true for all presented results; thus, we omit the error bars in the other tables for the sake of readability.

**UCM dataset** We observe in Table 2 that Algorithm 1 can compress the VGG11, VGG16 and ViT-16b networks equally well as the non-regularized low-rank compression and achieves the first goal of high compression values of up to $94\%$ reduction of trainable parameters. Furthermore, the clean accuracy is similar to the non-compressed baseline architecture; thus, we achieve the second goal of (almost) loss-less compression. Noting the adversarial accuracy results under the $\ell^2$-FGSM, Jitter, and Mixup attacks with various attack strengths $\epsilon$, we observe that across all tests, the regularized low-rank network of Algorithm 1 significantly outperforms the non-regularized low-rank network. For the $\ell^2$-FGSM attack, our method is able to recover the adversarial accuracy of the baseline network. For Mixup, the regularization almost doubles the baseline accuracy for VGG16. By targeting the condition number of the weights, which gives a bound on the *relative* growth of the loss w.r.t. the size of the input, we postulate that the large improvement could be attributed to the improved robustness against the scale invariance attack [27, Section 3.3] included in Mixup. We refer the reader to Appendix B.1.4 for a precise definition of the Mixup attack featuring scale invariance. However, this hypothesis was not further explored and is delayed to a future work. Finally, we are able to recover half of the lost accuracy in the Jitter attack. Overall, we achieved the third goal of significantly increasing adversarial robustness of the compressed networks. We refer to Table 9 for the used values of $\beta$ and Appendix A.1 for extended numerical results.

**Cifar10 dataset** We repeat the methodology of the UCM dataset for Cifar10, and observe similar computational results in Table 2. Furthermore, we compare our method in Table 4 to several methods of the recent literature, see Section 3. We compare the adversarial accuracy under the $\ell^1$-FGSM attack, see Appendix B.1.2 for details, for consistency with the literature results. We find that our proposed method achieves the highest adversarial validation accuracy for all attack strengths $\epsilon$, even surpassing the baseline adversarial accuracy. Additionally, we find an at least 15% higher compression ratio with RobustDLRT than the second best compression method, CondLR [35]. A similar experiment for the Projected Gradient Descent (PGD) attack [30] is given in Appendix A.2.

**ImageNet1k dataset** Finally we repeat the methodology for the ImageNet1k dataset, using the ViT-32l vision transformer trained from an ImageNet21k checkpoint, and report the results in Table 3. The hyperparameters are obtained by

Table 4: Comparison to literature on CIFAR10 with VGG16 under the $\ell^1$-FGSM attack. The first three rows list the computed mean over 10 random initializations. The values of all other methods, given below the double rule, are taken from [35, Table 1]. RobustDLRT has higher adversarial accuracy at higher compression rates than all listed methods.

| Method | c.r. [%] | $\ell^1$-**FGSM**, $\epsilon$ | | | |
|---|---|---|---|---|---|
| | | 0.0 | 0.002 | 0.004 | 0.006 |
| Baseline | 0 | 89.83 | 78.61 | 64.66 | 53.71 |
| DLRT | 94.58 | 89.55 | 74.71 | 59.61 | 47.56 |
| RobustDLRT $\beta = 0.15$ | 94.35 | 89.35 | **78.72** | **66.02** | **54.15** |
| Cayley SGD [25] | 0 | 89.62 | 74.46 | 58.16 | 45.29 |
| Projected SGD [1] | 0 | 89.70 | 74.55 | 58.32 | 45.74 |
| CondLR [35] $\tau = 0.5$ | 50 | 89.97 | 72.25 | 60.19 | 50.17 |
| CondLR [35] $\tau = 0.5$ | 80 | 89.33 | 68.23 | 48.54 | 36.66 |
| LoRA [17] | 50 | 89.97 | 67.71 | 48.86 | 38.49 |
| LoRA [17] | 80 | 88.10 | 64.24 | 42.66 | 29.90 |
| SVD prune [51] | 50 | 89.92 | 67.30 | 47.77 | 36.98 |
| SVD prune [51] | 80 | 87.99 | 63.57 | 42.06 | 29.27 |

an initial sweep and reported in Tables 8 and 9. RobustDLRT consistently yields higher Top-1/Top-5 accuracy across $\ell^2$-FGSM and Jitter attacks than DLRT, with especially pronounced gains at larger perturbations (e.g., $+9$ points in Top-1 accuracy under $\ell^2$-FGSM $\epsilon = 0.3$). These trends are consistent with our ViT experiments in Table 2, demonstrating that adversarial regularization enhances robustness without compromising scalability. We benchmark the training runtime of one ImageNet epoch on an A100 80GB GPU. DLRT requires 26m 07s, while RobustDLRT (with the regularizer) requires 27m 51s, corresponding to an overhead of approximately 3%. This overhead can likely be reduced with further implementation optimizations, indicating that our approach is computationally scalable.

**Black-box attacks** We investigate the scenario where an attacker has knowledge of the used model architecture, but not of the low-rank compression. We use the Imagenet-1k pretrained VGG16 and VGG11 and re-train it with Algorithm 1 and baseline training on the UCM data using the same training hyperparameters. Then we generate adversarial examples with the baseline network and evaluate the performance on the low-rank network with and without regularization. The results are given in Table 5. In this scenario, the weights from low-rank training, being sufficiently far away from the baseline, provide an effective defense against the attack. Further, the proposed regularization significantly improves the adversarial robustness when compared to the unregularized low-rank network. Even for extreme attacks with $\epsilon = 1$, the regularized network achieves $84.76\%$ and $87.33\%$ accuracy for VGG16 and VGG11 respectively.

**Adversarial Training** We evaluate the performance of low-rank training for VGG16 on the UCM dataset using adversarial training. Following [13], we use the $\ell^2$-FGSM attack for different values of $\epsilon$ and train on both 50% clean and attacked images per batch. The results reported in Table 6 illustrate that RobustDLRT is both compatible with and able to benefit from adversarial training. DLRT without regularization benefits from adversarial training, but exhibits a clear margin to RobustDLRT. Additionally, RobustDLRT is able to approximately match the non-compressed baseline.

Table 5: UCM dataset – Black-box attack. Adversarial images with the $\ell^2$-FGSM attack are generated by the baseline network for different values of $\epsilon$. The baseline, DLRT ($\beta = 0$), and RobustDLRT ($\beta = 0.075$) networks are then evaluated on these images. Regularized low-rank compression achieves high adversarial accuracy, even under strong attacks.

Table 6: UCM dataset – Adversarial Training. VGG16 is trained on 50% clean images and 50% images attacked with $\ell^2$-FGSM for various $\epsilon$. The displayed numbers are the mean of 5 repeated runs. RobustDLRT ($\beta = 0.075$) is superior to DLRT ($\beta = 0$) and is able to approximately match the non-compressed baseline.

| | Method | c.r. [%] | $\ell^2$-FGSM, $\epsilon$ | | | | | |
|---|---|---|---|---|---|---|---|---|
| | | | 0.05 | 0.1 | 0.25 | 0.5 | 0.75 | 1.0 |
| VGG16 | Baseline | 0.0 | 86.71 | 76.40 | 48.76 | 39.33 | 35.23 | 33.23 |
| | $\beta = 0$ | 95.30 | 93.03 | 91.81 | 88.09 | 83.14 | 78.95 | 76.00 |
| | $\beta = 0.05$ | 95.15 | 92.66 | 92.47 | 91.33 | 88.76 | 86.85 | 84.76 |
| VGG11 | Baseline | 0.0 | 89.93 | 78.66 | 60.76 | 45.23 | 38.38 | 35.52 |
| | $\beta = 0$ | 95.82 | 92.76 | 91.81 | 88.25 | 84.09 | 80.57 | 77.71 |
| | $\beta = 0.05$ | 96.12 | 92.95 | 92.66 | 92.00 | 91.04 | 88.66 | 87.33 |

| Method | c.r. [%] | $\ell^2$-FGSM, $\epsilon$ | | | | |
|---|---|---|---|---|---|---|
| | | 0.0 | 0.1 | 0.5 | 0.75 | 1.0 |
| Baseline | 0.0 | 92.61 | 91.91 | 91.90 | 89.61 | 89.91 |
| $\beta = 0$ | 94.46 | 92.55 | 91.91 | 87.98 | 85.37 | 82.96 |
| $\beta = 0.075$ | 94.19 | 92.49 | 92.49 | 90.98 | 89.56 | 89.42 |

## 7 Conclusion

RobustDLRT enables highly compressed neural networks with strong adversarial robustness by controlling the spectral properties of low-rank factors. The method is efficient, rank-adaptive, and yields an up to 94% parameter reduction across a diverse suite of models and datasets. The method achieves competitive accuracy, even for strong adversarial attacks, surpassing the current literature results by a significant margin. Therefore, we conclude the proposed method scores well in the combined metric of compression, accuracy and adversarial robustness.

The accomplished high compression and adversarial robustness advance computer vision models and enable broader applications on resource-constrained edge devices. These achievements also enhance energy efficiency and trustworthiness, positively impacting society. The regularization and condition number bounds further improve interpretability, which is crucial for transparency and accountability in critical decision-making when applying the proposed methods.

## Acknowledgments and Disclosure of Funding

This manuscript has been authored by UT-Battelle, LLC under Contract No. DE-AC05-00OR22725 with the U.S. Department of Energy. The United States Government retains and the publisher, by accepting the article for publication, acknowledges that the United States Government retains a non-exclusive, paid-up, irrevocable, world-wide license to publish or reproduce the published form of this manuscript, or allow others to do so, for United States Government purposes. The Department of Energy will provide public access to these results of federally sponsored research in accordance with the DOE Public Access Plan(http://energy.gov/downloads/doe-public-access-plan).

This material is based upon work supported by the Laboratory Directed Research and Development Program of Oak Ridge National Laboratory (ORNL), managed by UT-Battelle, LLC for the U.S. Department of Energy under Contract No. De-AC05-00OR22725.

S. Schotthöfer, H. L. Yang, and S. Schnake were supported by the Artificial Intelligence Initiative of the Laboratory Directed Research and Development Program of Oak Ridge National Laboratory (ORNL), managed by UT-Battelle, LLC for the U.S. Department of Energy under Contract No. De-AC05-00OR22725.

This research used resources of the Compute and Data Environment for Science (CADES) at the Oak Ridge National Laboratory, which is supported by the Office of Science of the U.S. Department of Energy under Contract No. DE-AC05-00OR22725.

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

# A Additional Numerical Results

## A.1 UCM Dataset

The numerical results for the whitebox $\ell^2$-FGSM, Jitter, and Mixup adversarial attacks on the VGG16 and VGG11 architectures can be found in Figure 4, Figure 5, and Figure 6. The regularizer confidently increases the adversarial validation accuracy of the networks.

In Table 10, we observe that the regularizer $\mathcal{R}(W)$ applied to the full weight matrices (and flattened tensors) $W$ in baseline format is able to increase the adversarial robustness of the baseline network in the UCM/VGG16 test case. However, the increased adversarial robustness comes at the expense of some of the clean validation accuracy.

## A.2 Cifar10 Dataset

We run the same experiment in Table 4 but with the $\ell^2$-PGD attack, which is an iterative version of $\ell^2$-FGSM with an random perturbation of the input image as the initial condition [30]. Overall, we see that RobustDLRT is competitive with the other robustness-improving methods when the compression rate is taken into account.

Table 7: Comparison to literature on CIFAR10 with VGG16 under the $\ell^2$-PGD attack. The first three rows list the computed mean over 10 random initializations. The values of all other methods, given below the double rule, are taken from [35, Table 5]. RobustDLRT has competitive adversarial accuracy to all methods with a compression rate $\geq 80\%$.

| Method | c.r. [%] | $\ell^2$-PGD, $\epsilon$ | | | | | | | |
|---|---|---|---|---|---|---|---|---|---|
| | | 0.0 | 0.1 | 0.13 | 0.16 | 0.2 | 0.23 | 0.26 | 0.3 |
| RobustDLRT $\beta = 0.15$ | 94.18 | 88.80 | 62.58 | 53.47 | 44.95 | 34.75 | 28.33 | 22.64 | 16.59 |
| DLRT | 94.53 | 88.58 | 59.34 | 50.06 | 41.50 | 31.82 | 25.67 | 20.48 | 15.04 |
| Baseline | 0 | 90.48 | 63.01 | 54.66 | 47.87 | 40.77 | 36.75 | 33.51 | 29.93 |
| Cayley SGD [25] | 0 | 89.62 | 67.68 | 59.38 | 51.09 | 40.87 | 34.46 | 29.21 | 23.62 |
| Projected SGD [1] | 0 | 89.70 | 67.64 | 59.25 | 51.06 | 40.86 | 34.51 | 29.19 | 23.64 |
| CondLR [35] $\tau = 0.1$ | 50 | 90.93 | 67.03 | 62.08 | 59.15 | 56.92 | 55.96 | 55.28 | 54.58 |
| CondLR [35] $\tau = 0.5$ | 50 | 89.97 | 64.84 | 60.25 | 57.75 | 56.03 | 55.21 | 54.75 | 54.25 |
| CondLR [35] $\tau = 0.1$ | 80 | 90.48 | 61.00 | 50.84 | 42.19 | 33.70 | 29.44 | 26.55 | 23.97 |
| CondLR [35] $\tau = 0.5$ | 80 | 89.33 | 57.45 | 46.35 | 37.20 | 28.30 | 23.82 | 20.65 | 17.84 |
| LoRA [17] | 50 | 89.97 | 55.74 | 45.11 | 36.86 | 29.62 | 26.28 | 24.02 | 21.84 |
| LoRA [17] | 80 | 88.10 | 51.40 | 39.70 | 30.12 | 20.97 | 16.29 | 13.15 | 10.37 |
| SVD prune [51] | 50 | 89.92 | 54.87 | 43.85 | 35.23 | 27.95 | 24.38 | 22.06 | 19.94 |
| SVD prune [51] | 80 | 87.99 | 50.64 | 39.06 | 29.57 | 20.16 | 15.49 | 12.22 | 9.57 |

# B Details to the numerical experiments of this work

## B.1 Recap of adversarial attacks

In the following we provide the defintions of the used adversarial attacks. We use the implementation of [50] for the $\ell^2$-FGSM, Jitter, and Mixup attack. For the $\ell^1$-FGSM attack, we use the implementation of `https://github.com/COMPiLELab/CondLR`.

### B.1.1 $\ell^2$-FGSM attack

The Fast Gradient Sign Method (FGSM)[21] is a single-step adversarial attack that perturbs an input in the direction of the gradient of the loss with respect to the input. Given a neural network classifier $f_\theta$ with parameters $\theta$, an input $x$, and its corresponding label $y$, the attack optimizes the cross-entropy loss $\mathcal{L}_{\text{CE}}(f_\theta(x), y)$ by modifying $x$ along the gradient's sign. The adversarial example is computed

as:

$$x' = x + \alpha \cdot \frac{\nabla_x \mathcal{L}_{\text{CE}}(f_\theta(x), y)}{\|\nabla_x \mathcal{L}_{\text{CE}}(f_\theta(x), y)\|_\infty}, \tag{12}$$

where $\alpha$ controls the perturbation magnitude. To ensure the perturbation remains bounded, the difference $x' - x$ is clamped by an $\epsilon$ bound, i.e.,

$$x' = x + \max(-\epsilon, \min(x' - x, \epsilon)). \tag{13}$$

In this work we fix $\alpha = \epsilon$. The attack can be iterated to increase its strength.

### B.1.2  $\ell^1$-FGSM attack

The $\ell^1$-FGSM attack [44] is used in the reference work of [35] and uses the same workflow as (B.1.1), where (12) is changed to

$$x' = x + \alpha \cdot \frac{\text{sign}(\nabla_x \mathcal{L}_{\text{CE}}(f_\theta(x), y))}{\Sigma}, \tag{14}$$

where $\Sigma$ denotes the standard deviation of the data-points in the training data-set and the sign of the gradient matrix is taken element wise.

### B.1.3  Jitter attack

The Jitter attack [39] is an adversarial attack that perturbs an input by modifying the softmax-normalized output of the model with random noise before computing the loss. Given a neural network classifier $f_\theta$ with parameters $\theta$, an input $x$, and its corresponding label $y$, the attack first computes the network output $z = f_\theta(x)$ and normalizes it using the $\ell^\infty$ norm:

$$\hat{z} = \text{Softmax}\left(\frac{s \cdot z}{\|z\|_\infty}\right), \tag{15}$$

where $s$ is a scaling factor. A random noise term $\eta \sim \mathcal{N}(0, \sigma^2)$ is added to $\hat{z}$, i.e.,

$$\tilde{z} = \hat{z} + \sigma \cdot \eta. \tag{16}$$

The attack loss function is a mean squared error between perturbed input and target, given by

$$\mathcal{L} = \|\tilde{z} - y\|_2^2. \tag{17}$$

The adversarial example is then computed using the gradient of $\mathcal{L}$ with respect to $x$:

$$x' = x + \alpha \cdot \frac{\nabla_x \mathcal{L}}{\|\nabla_x \mathcal{L}\|_\infty}. \tag{18}$$

To ensure the perturbation remains bounded, the modification $x' - x$ is clamped within an $\epsilon$ bound:

$$x' = x + \max(-\epsilon, \min(x' - x, \epsilon)). \tag{19}$$

In this work, we fix $\alpha = \epsilon$ and set $\sigma = 0.1$. The Jitter attack can be performed iteratively. Then, for each but the first iteration $k$, the attack loss is normalized by the perturbation of the input image,

$$\mathcal{L} = \frac{\|\tilde{z} - y\|_2^2}{\|x - x_k'\|_\infty}, \qquad k > 0 \tag{20}$$

In this work, we use 5 iterations of the Jitter attack for each image.

### B.1.4  Mixup attack

The Mixup attack [49] is an adversarial attack that generates adversarial samples that share similar feature representations with an given virtual example. Inspired by the Mixup data augmentation technique, this attack aims to create adversarial examples that maintain characteristics of both the original sample and its adversarial counterpart. Given a neural network classifier $f_\theta$ with parameters $\theta$, an input $x$, and its corresponding label $y$, the attack first computes a linear combination of cross-entropy and negative KL-divergence loss,

$$\mathcal{L}_{\text{mixup}} = \beta \sum_{k=1}^{5} \mathcal{L}_{\text{CE}}\left(f_\theta\left(\frac{x}{2^k}\right), y\right) - \mathcal{L}_{\text{KL}} \tag{21}$$

Table 8: Training hyperparameters for the UCM, Cifar10, and ImageNet Benchmarks. The first set hyperparameters apply to both DLRT and baseline training, and we train DLRT with the same hyperparameters as the full-rank baseline models. The second set of hyper-parameters is specific to DLRT. The DLRT hyperparameters are selected by an initial parameter sweep. We choose the DLRT truncation tolerance relative to the Frobenius norm of $\widehat{S}$, i.e. $\vartheta = \tau \|\widehat{S}\|_F$, as suggested in [38].

| Hyperparameter | VGG16 | VGG11 | ViT16b | ViT32l |
|---|---|---|---|---|
| Batch Size (UCM) | 16 | 16 | 16 | n.a. |
| Batch Size (Cifar10) | 128 | 128 | 128 | n.a. |
| Batch Size (ImageNet) | n.a. | n.a. | n.a. | 256 |
| Learning Rate | 0.001 | 0.001 | 0.001 | 0.001 |
| Number of Epochs | 20 | 20 | 5 | 10 |
| L2 regularization | 0 | 0 | 0.001 | 0.0001 |
| Optimizer | AdamW | AdamW | AdamW | AdamW |
| DLRT rel. truncation tolerance $\tau$ | 0.1 | 0.05 | 0.08 | 0.013 |
| Coefficient Steps $s_*$ | 10 | 10 | 10 | 75 |
| Initial Rank | 150 | 150 | 150 | 200 |
| Parameters | 138M | 132M | 86M | 304M |

$$\delta = \alpha \cdot \frac{\nabla_x \mathcal{L}_{\text{CE}}(f_\theta(x), y)}{\|\nabla_x \mathcal{L}_{\text{CE}}(f_\theta(x), y)\|_\infty}. \tag{22}$$

Equation (21) features a scale invariance attack applied to the loss [27, Section 3.3].

The final adversarial example is computed as a convex combination of the original input and its perturbed version:

$$x' = \lambda x + (1 - \lambda)(x + \delta), \tag{23}$$

where $\lambda \sim \text{Beta}(\beta, \beta)$ is sampled from a Beta distribution with hyperparameter $\beta$, controlling the interpolation between clean and perturbed inputs. The perturbation is further constrained within an $\epsilon$-ball to ensure bounded adversarial modifications:

$$x' = x + \max(-\epsilon, \min(x' - x, \epsilon)). \tag{24}$$

In this work, we fix $\alpha = 1$ and set $\beta = 10^{-3}$. The attack can be iterated to increase its effectiveness, refining the adversarial perturbation at each step. We use 5 iterations of the Mixup Attack for each image.

## B.2 Network architecture and training details

In this paper, we use the pytorch implementation and take pretrained weights from the imagenet1k dataset as initialization. The data-loaded randomly samples a batch for each batch-update which is the only source of randomness in our training setup. Below is an overview of the used network architectures

- VGG16 is a deep convolutional neural network architecture that consists of 16 layers, including 13 convolutional layers and 3 fully connected layers.
- VGG11 is a convolutional neural network architecture similar to VGG16 but with fewer layers, consisting of 11 layers: 8 convolutional layers and 3 fully connected layers. It follows the same design principle as VGG16, using small 3×3 convolution filters and 2×2 max-pooling layers.
- ViT16b is a Vision Transformer with 16x16 patch size, a deep learning architecture that leverages transformer models for image classification tasks.
- ViT32l is a Vision Transformer with 32x32 patch size, a deep learning architecture that leverages transformer models for image classification tasks. We use the Imagenet21k weights from the huggingface endpoint google/vit-large-patch32-224-in21k as weight initialization.

The full training setup is described in Table 8. We train DLRT with the same hyperparameters as the full-rank baseline models. It is known [37] that DLRT methods are robust w.r.t. common

Table 9: Overview of the $\beta$ for best performing regularization strength for RobustDLRT of Table 2.

| Architecture | UCM Dataset | | | Cifar10 Dataset | | | ImageNet Dataset | | |
| | FGSM | Jitter | Mixup | FGSM | Jitter | Mixup | FGSM | Jitter | Mixup |
|---|---|---|---|---|---|---|---|---|---|
| VGG16 | 0.075 | 0.2 | 0.15 | 0.05 | 0.05 | 0.05 | n.a. | n.a. | n.a. |
| VGG11 | 0.1 | 0.05 | 0.15 | 0.15 | 0.05 | 0.2 | n.a. | n.a. | n.a. |
| ViT16b | 0.1 | 0.15 | 0.15 | 0.01 | 0.01 | 0.05 | n.a. | n.a. | n.a. |
| ViT32l | n.a. | n.a. | n.a. | n.a. | n.a. | n.a. | 0.075 | 0.075 | 0.075 |

Table 10: UCM Data, VGG16, baseline training. Data is averaged over 10 stochastic training runs. The regularizer is able to increase the adversarial robustness of the baseline training network, at the cost of some reduction of its clean validation accuracy. The provided results are averaged over 5 iterations.

| $\beta$ | Acc [%] under the $\ell^2$-FGSM attack with $\epsilon$ | | | | | | | | | |
| | 0 | 0.01 | 0.025 | 0.05 | 0.075 | 0.1 | 0.2 | 0.3 | 0.4 | 0.5 |
|---|---|---|---|---|---|---|---|---|---|---|
| 0 | 92.40 | 91.72 | 90.65 | 86.71 | 81.32 | 76.40 | 64.52 | 54.96 | 49.38 | 45.14 |
| 0.0001 | 91.69 | 91.69 | 91.10 | 87.73 | 83.14 | 78.43 | 63.21 | 53.31 | 47.18 | 42.99 |
| 0.001 | 88.81 | 88.78 | 87.90 | 84.40 | 80.00 | 76.34 | 62.61 | 53.77 | 48.09 | 44.38 |
| 0.01 | 88.22 | 88.19 | 87.12 | 82.78 | 77.52 | 72.72 | 58.32 | 48.89 | 42.83 | 38.61 |
| 0.05 | 90.45 | 90.43 | 89.63 | 87.23 | 84.11 | 80.55 | 68.66 | 59.29 | 52.62 | 46.61 |
| 0.1 | 92.51 | 92.51 | 92.11 | 90.45 | 88.43 | 86.32 | 76.91 | 68.01 | 61.29 | 55.52 |
| 0.2 | 89.20 | 89.18 | 88.85 | 86.66 | 84.36 | 81.96 | 73.25 | 65.20 | 58.61 | 53.29 |

hyperparameters as learning rate, and batch-size, and initial rank. The truncation tolerance $\tau$ is chosen between $0.05$ and $0.1$ per an initial parameter study. These values are good default values, as per recent literature [36, 42]. In general, there is a trade-off between target compression ratio and accuracy, as illustrated e.g. in [38] for matrix-valued and [42] for tensor-valued (CNN) layers.

## B.3 UCM Test Case

The University of California, Merced (UCM) Land Use Dataset is a benchmark dataset in remote sensing and computer vision, introduced in [52]. It comprises 2,100 high-resolution aerial RGB images, each measuring 256×256 pixels, categorized into 21 land use classes with 100 images per class. The images were manually extracted from the USGS National Map Urban Area Imagery collection, covering various urban areas across the United States. The dataset contains images with spatial resolution approximately 0.3 meters per pixel (equivalent to 1 foot), providing detailed visual information suitable for fine-grained scene classification tasks.

We normalize the training and validation data with mean $[0.485, 0.456, 0.406]$ and standard deviation $[0.229, 0.224, 0.225]$ for the rgb image channels. The convolutional neural neural networks used in this work are applied to the original $256 \times 256$ image size. The vision transformer data-pipeline resizes the image to a resolution of $224 \times 224$ pixels. The adversarial attacks for this dataset are performed on the resized images.

## B.4 Cifar10

The Cifar10 dataset consists of 10 classes, with a total of 60000 rgb images with a resolution of $32 \times 32$ pixels.

We use standard data augmentation techniques. That is, for CIFAR10, we augment the training data set by a random horizontal flip of the image, followed by a normalization using mean $[0.4914, 0.4822, 0.4465]$ and std. dev. $[0.2470, 0.2435, 0.2616]$. The test data set is only normalized. The convolutional neural neural networks used in this work are applied to the original $32 \times 32$ image size. The vision transformer data-pipeline resizes the image to a resolution of $224 \times 224$ pixels. The adversarial attacks for this dataset are performed on the resized images.

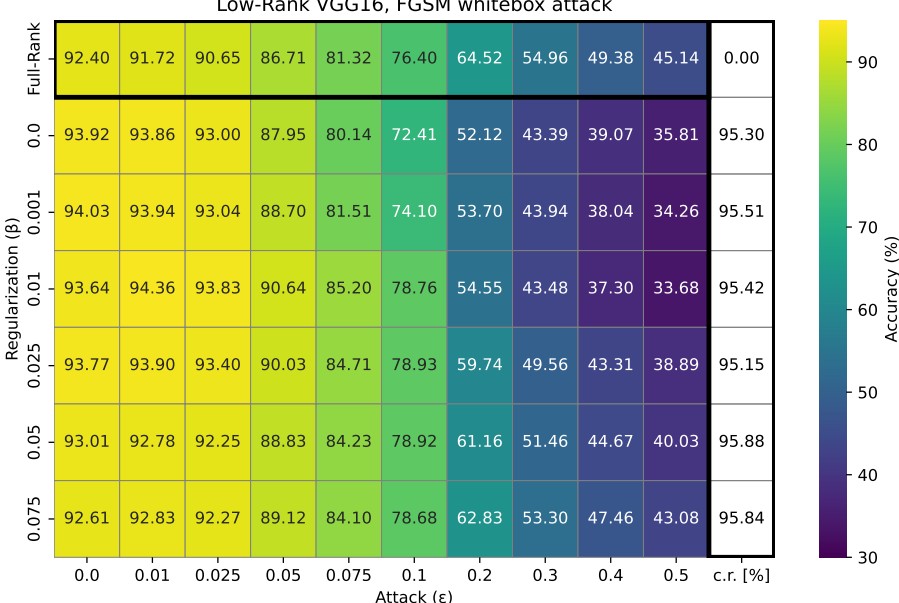

Figure 4: UCM Dataset, VGG16 clean and adversarial accuracy under the FGSM attack. Data is averaged over 10 stochastic training runs. The top row displays the full baseline network with $0\%$ c.r. and the matrix below displays the low-rank and regularized networks trained with Algorithm 1. All numbers display the mean of $10$ randomized training runs, where the randomness stems from shuffled batches. The initial condition of all runs is given by Imagenet-1k pretrained weights. The regularized low-rank networks with $\beta = 0.075$ are able to recover the adversarial robustness of the baseline training while compressed by 95.84%. Results for VGG11 and Vit16b are similar.

## B.5 ImageNet-1k

The ImageNet dataset consists of 1000 classes and over 1.2 million RGB training images, with a standard resolution of $224 \times 224$ pixels. We follow the standard data augmentation pipeline for ImageNet, which includes a random resized crop to $224 \times 224$, and normalization using mean $[0.5, 0.5, 0.5]$ and standard deviation $[0.5, 0.5, 0.5]$. The test set is only resized and center-cropped to $224 \times 224$, followed by normalization. Adversarial attacks are generated on the normalized, resized images.

## B.6 Computational hardware

All experiments in this paper are computed using workstation GPUs. Each training run used a single GPU. Specifically, we have used 5 NVIDIA RTX A6000, 3 NVIDIA RTX 4090, and 8 NVIDIA A-100 80G.

The estimated time for one experimental run depends mainly on the data-set size and neural network architecture. For training, generation of adversarial examples and validation testing we estimate 30 minutes on one GPU for one run.

## C  Proofs

To facilitate the proofs, we remark the definition of L-continuity: A function $f(x)$ is Lipschitz continuous on a domain $D$ if there exists a constant $L \geq 0$ such that for all $x, y \in D$,

$$\|f(x) - f(y)\| \leq L\|x - y\|.$$

The smallest such $L$ is called the Lipschitz constant.

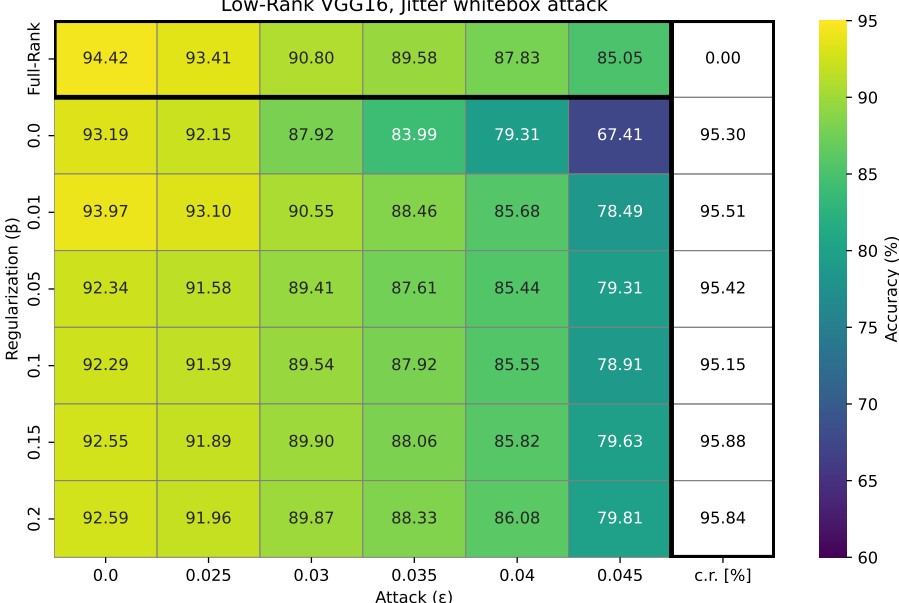

Figure 5: UCM Dataset, VGG16 clean and adversarial accuracy under the Jitter attack. Data is averaged over 10 stochastic training runs. The top row displays the full baseline network with $0\%$ c.r. and the matrix below displays the low-rank and regularized networks trained with Algorithm 1. All numbers display the mean of $10$ randomized training runs, where the randomness stems from shuffled batches. The initial condition of all runs is given by Imagenet-1k pretrained weights. The regularized low-rank networks are able to recover most of the adversarial robustness of the baseline network. Results for VGG11 and Vit16b are similar.

For the following proofs, let

$$(A, B) = \text{trace}(B^\top A) = \sum_{ij} A_{ij} B_{ij}$$

be the Frobenius inner product that induces the norm $\|A\| = \sqrt{(A, A)}$. By the cyclic property of the trace, we have

$$(AB, CD) = (B, CDA^\top) = (C^\top AB, D). \tag{25}$$

for matrices $A$, $B$, $C$, and $D$ of appropriate size.

*Proof of* (3). We calculate

$$
\begin{aligned}
\mathcal{R}(S)^2 &= (S^\top S - \alpha_S^2 I, S^\top S - \alpha_S^2 I) \\
&= \|S^\top S\|^2 - 2\alpha_S^2 (S^\top S, I) + \alpha_S^4 (I, I) \\
&= \|S^\top S\|^2 - \tfrac{1}{r}\|S\|^4 \\
&= \sum_{i=1}^{r} \varsigma_i(S^\top S)^2 - \frac{1}{r}\left(\sum_{i=1}^{r} \varsigma_i(S)^2\right)^2 \\
&= r\left(\frac{1}{r}\sum_{i=1}^{r} \varsigma_i(S^\top S)^2 - \left(\frac{1}{r}\sum_{i=1}^{r} \varsigma_i(S)^2\right)^2\right)
\end{aligned} \tag{26}
$$

Since $S^\top S$ is symmetric positive semi-definite, $\varsigma_i(S^\top S) = \varsigma_i(S)^2$. Applying this substitution yields (3). The proof is complete. $\square$

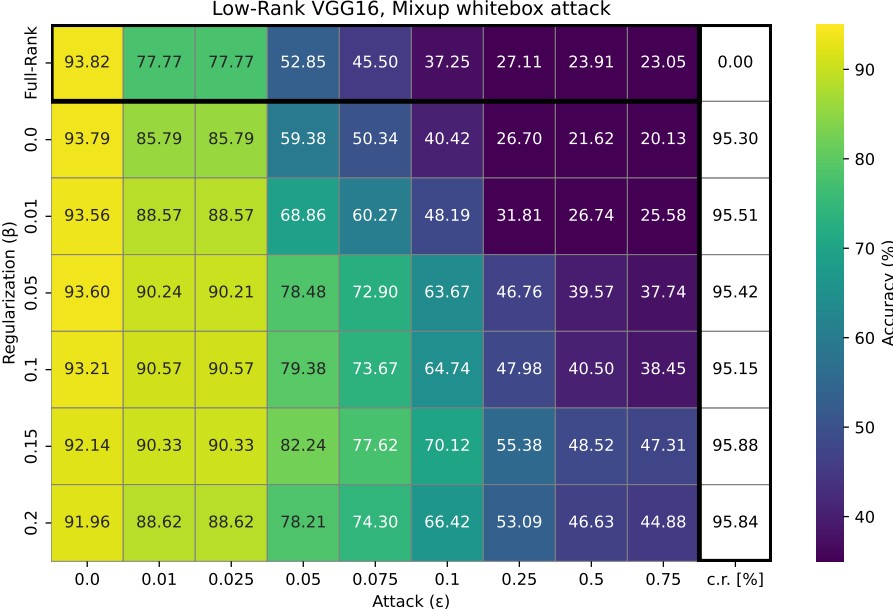

Figure 6: UCM Dataset, VGG16 clean and adversarial accuracy under the Mixup attack. Data is averaged over 10 stochastic training runs. The top row displays the full baseline network with $0\%$ c.r. and the matrix below displays the low-rank and regularized networks trained with Algorithm 1. All numbers display the mean of $10$ randomized training runs, where the randomness stems from shuffled batches. The initial condition of all runs is given by Imagenet-1k pretrained weights. The regularized low-rank networks almost double the adversarial accuracy of the baseline network at $95.84\%$ compression rate. Results for VGG11 and Vit16b are similar.

*Proof of Proposition 1.* Given $S \in \mathbb{R}^{r \times r}$, the Fréchet derivative for $\mathcal{Q} = \mathcal{R}^2$ at $S$ is a linear operator $Z \to \nabla \mathcal{Q}(S)[Z]$ for $Z \in \mathbb{R}^{r \times r}$. Denote $W_S = S^\top S - \alpha_S^2 I$ which is symmetric. Since $\mathcal{Q}$ is an inner product, we calculate $\nabla \mathcal{Q}(S)[Z]$ as

$$
\begin{aligned}
\tfrac{1}{2}\nabla \mathcal{Q}(S)[Z] &= (W_S, Z^\top S + S^\top Z - \tfrac{2}{r}(S, Z)I) \\
&= (W_S, Z^\top S) + (W_S, S^\top Z) - \tfrac{2}{r}(S, Z)(W_S, I) \\
&= (SW_S^\top, Z) + (SW_S, Z) - \tfrac{2}{r}(S, Z)(W_S, I) \\
&= 2(S(S^\top S - \alpha_S^2 I), Z) - \tfrac{2}{r}(S, Z)(S^\top S - \alpha_S^2 I, I).
\end{aligned}
\tag{27}
$$

Note by definition of $\alpha_S^2$,

$$
(S^\top S - \alpha_S^2 I, I) = \|S\|^2 - \alpha_S^2 \|I\|^2 = 0.
\tag{28}
$$

Hence

$$
\nabla \mathcal{Q}(S) = 4S(S^\top S - \alpha_S^2 I).
\tag{29}
$$

Since $\mathcal{R}^2 = \mathcal{Q}$, therefore

$$
\nabla \mathcal{R}(S) = \tfrac{\nabla \mathcal{Q}(S)}{2\mathcal{R}(S)}.
\tag{30}
$$

The desired estimate follows. The proof is complete. □

*Proof of Proposition 2.* From (26) there holds

$$
\frac{1}{r}\mathcal{R}(S)^2 = \frac{1}{r}\sum_{i=1}^{r}\varsigma_i(S^\top S)^2 - \left(\frac{1}{r}\sum_{i=1}^{r}\varsigma_i(S^\top S)\right)^2.
\tag{31}
$$

From (31), $\frac{1}{r}\mathcal{R}(S)^2$ is the variance of the sequence $\{\varsigma_i(S^\top S)\}_{i=1}^r$. The Von Szokefalvi Nagy inequality [33] bounds the variance of a finite sequence of numbers below by the range of the sequence (see [41]). Applied to (31), this yields

$$\frac{1}{r}\mathcal{R}(S)^2 \geq \frac{(\varsigma_1(S^\top S) - \varsigma_r(S^\top S))^2}{2r} = \frac{(\varsigma_1(S)^2 - \varsigma_r(S)^2)^2}{2r}. \tag{32}$$

Hence

$$\sqrt{2}\mathcal{R}(S) \geq \varsigma_1(S)^2 - \varsigma_r(S)^2. \tag{33}$$

An application of the Mean Value Theorem for logarithms (see [34, Proof of Theorem 2.2]), gives

$$\ln(\kappa(S)) \leq \frac{\varsigma_1(S)^2 - \varsigma_r(S)^2}{2\varsigma_r(S)^2}. \tag{34}$$

Combining (33) and (34) yields

$$\ln(\kappa(S)) \leq \frac{1}{\sqrt{2}\varsigma_r(S)^2}\mathcal{R}(S), \tag{35}$$

which, after exponentiation, yields (4). The proof is complete. $\qquad\square$

*Proof of Proposition 3.* Since $W$ is constant, we rewrite the dynamical system $\dot{S} + \beta\nabla\mathcal{R}(S) + S = W$ as

$$\frac{\mathrm{d}}{\mathrm{d}t}(S - W) + \beta\nabla\mathcal{R}(S) + (S - W) = 0. \tag{36}$$

Testing (36) by $S - W$ and rearranging yields

$$\tfrac{1}{2}\tfrac{\mathrm{d}}{\mathrm{d}t}\|S - W\|^2 + \beta(\nabla\mathcal{R}(S), S) + \|S - W\|^2 = \beta(\nabla\mathcal{R}(S), W). \tag{37}$$

We calculate $(\nabla\mathcal{R}(S), S)$. Note

$$(S(S^\top S - \alpha_S^2 I), S) = (S^\top S - \alpha_S^2 I, S^\top S)$$
$$= \|S^\top S\|^2 - \alpha_S^2(I, S^\top S) = \|S^\top S\|^2 - \tfrac{1}{r}\|S\|^4 = \mathcal{R}(S)^2, \tag{38}$$

where the last equality is due to (26). Hence

$$(\nabla\mathcal{R}(S), S) = \frac{2S(S^\top S - \alpha_S^2 I, S)}{\mathcal{R}(S)} = 2\mathcal{R}(S). \tag{39}$$

Using Hölder's inequality, the sub-multiplicative property of $\|\cdot\|$, and Young's inequality, we bound the right hand side of (37) by

$$\beta(\nabla\mathcal{R}(S), W) \leq 2\beta\frac{\|S(S^\top S - \alpha_S^2 I)\|}{\mathcal{R}(S)}\|W\| \leq 2\beta\|S\|\|W\|$$
$$\leq 2\beta(\|S - W\|\|W\| + \|W\|^2) \leq \tfrac{1}{2}\|S - W\|^2 + 2\beta(1 + 2\beta)\|W\|^2. \tag{40}$$

Applying (39) and (40) to (36) we obtain

$$\tfrac{1}{2}\tfrac{\mathrm{d}}{\mathrm{d}t}\|S - W\|^2 + 2\beta\mathcal{R}(S) + \tfrac{1}{2}\|S - W\|^2 \leq 2\beta(1 + 2\beta)\|W\|^2. \tag{41}$$

An application of Grönwall's inequality on $[0, t]$ yields

$$\tfrac{1}{2}\|S(t) - W\|^2 + 2\beta\int_0^t e^{\tau - t}\mathcal{R}(S(\tau))\,\mathrm{d}\tau = \tfrac{1}{2}e^{-t}\|S(0) - W\|^2 + 2(1 - e^{-t})\beta(1 + 2\beta)\|W\|^2. \tag{42}$$

The proof is complete. $\qquad\square$

