# OpenReview forum: "Dynamical Low-Rank Compression of Neural Networks with Robustness under Adversarial Attacks"
_NeurIPS.cc/2025/Conference — NeurIPS 2025 oral_

### Official Review · Reviewer_Gtfv · 2025-06-08

**Clarity:** 2
**Significance:** 2
**Originality:** 3
**Rating:** 4
**Confidence:** 4

**Summary:**

This paper provides a theortical derivation on the relationship between weight condition number and the model robustness. The derivation leads to a novel regularization on the condition number during the dynamic low-rank training process. The paper claims that the proposed method can mitigate the sensitivity of low-rank decomposed model to adversarial attacks without sacrificing accuracy on clean data.

**Questions:**

1. How does the impact of compression appear in the derivation of Sec. 4? Given a similar R, does model of different ranks share a similar condition number bound? If that's the case, why compressed model tends to be less robust as shown in the experiment?
2. As a follow up, experiments in Table 2 should be expanded to include additional results conducted with full rank trained model but regularized with the proposed regularization, so as to show the effectiveness of the regularization against baseline.

**Ethical Concerns:**

["NO or VERY MINOR ethics concerns only"]

**Final Justification:**

During the rebuttal, the author resolves the concerns on the interaction between robsutness and compression, the scalability of the proposed method, and effectiveness under adversarial training. I believe the paper has good merits to be accepted to the coference.

**Limitations:**

Yes

**Paper Formatting Concerns:**

No formatting concerns

**Quality:**

3

**Strengths And Weaknesses:**

## Strength
1. This paper provides a novel perspective of connecting model robustness to weight condition number and makes important observation in Fig 1 on the impact of low-rank training on the condition number
2. The paper proposes a novel and theortically sound regularization to regularize the condition number during low-rank training process

## Weakness
1. The paper claims to tackle efficiency and robustness together with the proposed method. However, the derivation in Sec. 4 only focus on the condition number's impact on robustness, but does not covers the impact of compression on the robustness or the impact of the regularization on the model comression rate.
2. This lack of clarity between the impact of regualrization and compression also appears in the experiment results. For example, Table 2 shows that RobustDLRT, which should have a lower condition number, still suffers from a lower robustness compared to baseline model.
3. The evaluation method is relatively weak in supporting the claim of the paper. Only weak adversarial attacks like FGSM is applied to show robustness improvement, which does not lead to a significant conclusion. Similarly, experiments are only applied on CIFAR-10, which lacks practical impact.

---

> ### Author Rebuttal · Authors · 2025-07-30
>
> We thank the reviewer for their constructive feedback.  We appreciate their assessment of the novelty of the method and the  soundness of the theory.
>
>
> 1. (Q Cross influence of regularization and compression)   From an analytical perspective, we only target the spectrum to enable a well-conditioned low-rank matrix.
>  However, we provide results on the effect of the regularizer on the compression rate in Figure 4,5,7 in the Appendix, which show a full result matrix of RobustDLRT-trained low-rank networks with different regularization strenghts $\beta$ and their compression rate (and adversarial accuracy under different attacks). It becomes apparent that the compression rate of the networks is very similar and does not deepened on the regularization strength.
>
>      We show the impact of compression on robustness in all Tables in the main manuscript, where it is apparent that the adversarial robustness of non-regularized, compressed networks suffers. It can be recovered by application of the proposed regularizer as we show in multiple test cases, e.g. Table 1-4. This is the main contribution of this work.
>
>
> 2. (Q Clarity of regularization vs compression)  We note that the case of Figure 1, where the DLRT spectrum is a widened version of the baseline spectrum and the regularizer straightens the spectrum, is not present across all layers.  In some layers, the spectrum of DLRT and robustDLRT is far larger than the baseline model; this fact could explain the lack of strong correlation between condition number and adversarial accuracy.  As an aside, a portion of the above comment is present in a footnote in the original work, but was not properly referenced in the caption of Figure 1.
>
>     Additionally, in the cases where the baseline is acceptable already, RobustDLRT shares similar adversarial accuracy.
>
>     We finally comment that creating a robust network will require several defenses combined, and this paper is to demonstrate the capabilities of the proposed method.
>
> 3. (Q Evaluation Method)  In Table 2, we evaluate the baseline, DLRT, and RobustDLRT models with the FGSM, Jitter, and Mixup adversarial attacks.  Jitter and Mixup, described with references in Sections A.1.3 and A.1.4 respectively, are both iterative and gradient-based adversarial attacks that are quite strong.
>
>     To eleviate the concerns of scalability we provide numerical results for the ImageNet-1k (1.2 Million images), where we compress the ViT-L32 vision transformer (304 Million parameters). We first conduct a hyperparameter sweep to determine standard hyperparameters as learning rate, weight-decay and batch size, which we report in Appendix C.
>
>     We compare then the baseline model, DLRT and the regularized RobustDLRT adversarial performance in the table below and observe similar trends as in the Vision Transformer experiments of Table 2 in the main manuscript.
>
>
> **Table: ImageNet, Top-1 Accuracy under various attacks**
>
> | Method      | c.r. [%] | Clean Acc. | FGSM ε=0.05 | FGSM ε=0.1 | FGSM ε=0.3 | Jitter ε=0.035 | Jitter ε=0.045 |
> |-------------|-----------|-------------|-------------|-------------|-------------|------------------|------------------|
> | Baseline    | 0         | 74.37       | 43.58       | 31.42       | 16.03       | 43.09           | 35.57           |
> | DLRT        | 58.02     | 72.27       | 42.70       | 30.32       | 15.47       | 43.98           | 38.44           |
> | RobustDLRT  | 57.98     | 72.25       | 43.17       | 35.11       | 25.24       | 48.22           | 43.51           |
>
> **Table: ImageNet, Top-5 Accuracy under various attacks**
>
> | Method                  | c.r. [%] | Clean Acc. | FGSM ε=0.05 | FGSM ε=0.1 | FGSM ε=0.3 | Jitter ε=0.035 | Jitter ε=0.045 |
> |-------------------------|-----------|-------------|-------------|-------------|-------------|------------------|------------------|
> | Baseline                | 0         | 92.20       | 73.75       | 63.42       | 43.41       | 78.24           | 74.96           |
> | DLRT                    | 58.02     | 90.06       | 70.43       | 60.90       | 40.58       | 74.49           | 71.31           |
> | RobustDLRT              | 57.98     | 90.03       | 71.58       | 62.82       | 50.65       | 77.35           | 75.14           |
>
>
>
> Finally we wish to remark that the slightly lower compression rate is expected since the hidden dimensions of ViT-L32 of 1024 is close to the number of ImageNet classes (1000), thus there is less redundancy in the model compared to other reported benchmarks.
>
>
>
> 4. (Q Impact of the compression in derivation of Sec. 4)
>     1) The compression of the model is largely achieved through the rank-augmentation and rank-truncation steps of Algorithm 1.  The regularizer influences the spectrum of the latent space map $\hat{S}$ which affects the truncation criterion  of line 251.   The regularizer may negatively affect compression by raising small singular values that would be truncated if not regularizer were used; however, we find in our experiments that the compression rate is not affected in practice.
>
>         We feel the two follow up questions is addressed in the response to your second question (Q) above.
>
> 5. (Q Full rank regularization) We have conducted this experiment in Table 8 in the original manuscript in the Appendix, where we explore the effects on regularization of a full-rank network, were we have observed: "The regularizer is able to increase the adversarial robustness of the baseline training network, at the
> cost of some reduction of its clean validation accuracy."
>
>     We wish to remark, that using $\mathcal{R}$ on the full-rank weight layer is computationally expensive, and that the regularizer is designed for low-rank networks.

---

> > ### Comment · Reviewer_Gtfv · 2025-08-03
> >
> > Thanks for the rebuttal. The rebuttal resolves my concern on the interaction between compression and robustness. The additional experiment on ImageNet is also greatly appreciated.
> >
> > Meanwhile, as I went through other reviews, I believe Reviewer HEm3 makes a valid point on the adversarially trained baseline. Adversarial training, as a go-to method for robutness improvement, should be considered to show the practical impact of the proposed method. I would suggest comparing the proposed regularization with adversarial training + compression or try if the proposed regularization can work together with adversarial training.

---

> > > ### Author Response · Authors · 2025-08-05
> > >
> > > We thank the reviewer for their comment and are happy that the imagenet results are appreciated.
> > >
> > > Below we address the concern about compatibility with adversarial training.
> > >
> > > **Adversarial Training**
> > >
> > > We evaluate VGG16 on UCM with the training hyperparameters and setup of Table 2 in our original manuscript using adversarial training. Following [1], we use the FGSM attack for different values of $\epsilon$ and  train on both 50% clean and attacked images per batch.
> > >
> > > The results below illustrate that RobustDLRT is able to benefit from adversarial training. DLRT without regularization benefits slightly from adversarial training, but exhibits a clear margin to RobustDLRT. RobustDLRT is able to approximately match the non-compressed baseline. RobustDLRT achieves approximately 93% compression rate. The displayed numbers are the mean of 5 repeated runs.
> > >
> > > [1] Explaining and Harnessing Adversarial Examples: Ian J. Goodfellow, Jonathon Shlens, Christian Szegedy
> > >
> > > |Epsilon | 0  | 0.1  | 0.5  | 0.75  | 1  |
> > > |-------|-------|-------|-------|-------|-------|
> > > Base| 92.61 | 91.91 | 91.90 | 89.61 | 89.91 |
> > > DLRT| 92.55 | 91.91 |  87.98  | 85.37 | 82.96 |
> > > RobustDLRT| 92.49  | 92.49 |  90.98 | 89.56 | 89.42 |
> > >
> > > We hope that these results can convince the reviewer of the practicability of the method.

---

> > > > ### Comment · Reviewer_Gtfv · 2025-08-05
> > > >
> > > > These additional results are helpful. I would like to increase my score accordingly. Please include these additional results in your final revision.

---

### Official Review · Reviewer_HEm3 · 2025-06-15

**Clarity:** 1
**Significance:** 3
**Originality:** 3
**Rating:** 4
**Confidence:** 3

**Summary:**

This paper addresses the challenge of preserving adversarial robustness during low-rank model compression. The authors begin with a preliminary analysis showing that the compression process can alter the condition number of the model, potentially impacting robustness. To address this, they introduce a robustness-oriented regularization technique that explicitly controls the condition number of the low-rank decomposition. Both theoretical analysis and empirical evaluations are presented to support the effectiveness of the proposed method.

**Questions:**

The results in Table 1 show that RobustDLRT can improve the performance under a large perturbation budget, where the baseline performs very badly. I am wondering whether you are using an adversarial trained model or just a normal trained model as the baseline.

**Ethical Concerns:**

["NO or VERY MINOR ethics concerns only"]

**Final Justification:**

In the rebuttal, the authors solved my major concern about the baseline problem. Therefore, I increase my rating.

**Limitations:**

Yes.

**Paper Formatting Concerns:**

No concerns.

**Quality:**

2

**Strengths And Weaknesses:**

# Strength #

* The research topic of this paper is practical. It is always a concern about the trustworthiness of the model during model compression. This paper tries to solve the challenge, which is beneficial to the real model application.

* The proposed method is easy but effective. The proposed method designs a regularization term about the condition number, which will not increase the training cost too much. In this case, the experimental results can outperform DLRT significantly, especially when the attack budget is large.

# Weakness #

* The outline of the paper is strange, making it hard to follow the logic of the paper. For instance, Section 2 shows some preliminary knowledge of low-rank compression. However, the related works are shown in the next section. I suggest to directly to the method part after the preliminary. Moreover, in Section 5, there is an individual subsection 5.1, but with all other contents in the main section.

* Some definitions in the paper are not clear.

-- The Lipschitz continuity should be explained in a formula.

-- In line 78, the definition of the condition number of the activation function is referred to in related works.

-- In equation (6), the definition of "orth( | )" is not clear.

* The method part is messy, and it is hard to discriminate which part is the proposed part. Most of the content in Section 5 is similar to FeDLRT [1], including the * Basis Augmentation * and * Latent Space Training *. As I understand, the major difference is the change in the gradient computation after introducing the regularization term.

* The baselines are not clear in the Experimental section. There should be a paragraph to explain the compared methods. I wonder which DLRT is applied for comparison. Besides, I expect FeDLRT is compared since the proposed regularization is applied to it.

* Typo: line 62, the compression of the network

In general, I expect the authors can reorganize the paper and elaborate on which part is from existing works and which is the innovation part of this paper.

[1] Schotthöfer, Steffen, and M. Paul Laiu. "Federated dynamical low-rank training with global loss convergence guarantees." arXiv preprint arXiv:2406.17887 (2024).

---

> ### Author Rebuttal · Authors · 2025-07-30
>
> We thank the reviewer for their constructive review. We appreciate that they found the proposed method practical, efficient and easy to implement.
>
>
> 1. (Q Lipschitz definition) We added the definition of Lipschitz continuity in the appendix. For completeness:  A function $ f(x) $ is **Lipschitz continuous** on a domain $ D $ if there exists a constant $ L \geq 0 $ such that for all $ x, y \in D $,
> $$
> \|f(x) - f(y)\| \leq L \|x - y\|.
> $$
> The smallest such $ L $ is called the **Lipschitz constant**.
>
>
> 2. (Q orth operator) As described in line 235 of the original manuscript We need to check all line numbers before resubmission,   \( Q=orth( Z )  \) is defined as the $Q$ factor in the QR-factorization, i.e, $QR=qr(Z)$.
>  Additionally, $[A|B]$ for $ A,B\in\mathbb{R}^{n\times r}$ concatenates $A,B$ in the second dimension, i.e. $[A|B]\in\mathbb{R}^{n\times 2r}$.
>
>
> 3. (Q method part) We wish to remark that the main contribution is the construction of the Regularizer $\mathcal{R}$ in Section 4. We state in line 153 that the method is compatible with any low-rank training scheme, e.g. [48, 30, 31, 29], that provides orthonormality of the basis $U,V$.
>
>     To facilitate the implementation we choose the scheme of [30] ("FeDLRT") as the backbone scheme. We do not claim invention of FeDLRT here, but incorporate our regularizer into it. The applicability of the regularizer to a wide range of low-rank training methods is an advantage, as a user does not need to change their implementation to use our regularizer.
>
>
> 4. (Q Baselines) We denote by "Baseline" the default training of the full model (not low-rank factorized) with the training hyperparameters specified in Appendix A2; this is stated in Section 6, Line 293.  DLRT and RobustDLRT use the two-step scheme of [30], which is described in detail in section 5. As denoted in the caption of Table 2,  RobustDLRT with regularization parameter $\beta=0$ equals DLRT. The same methods are used in all other experiments. All other compared methods are cited.
>
> 5. (Q Adversarial training) In all experiments presented in this paper, we train solely on clean data without employing adversarial training. This choice is intentional: the proposed method acts as a regularizer and is conceptually orthogonal to data-driven approaches like adversarial training. To clearly isolate and illustrate the effect of the regularizer, we evaluate it in the unaltered, clean-data training setting.
>
> 6. (Typos) We fixed the typos - Thank you for noticing!

---

> ### Comment · Reviewer_HEm3 · 2025-08-02
>
> Thanks for the clarification of the definition and the answers by the authors. However, I still have the following concerns.
>
> 1. **Clarification of Section 5**. While I understand the motivation to highlight the compatibility of the proposed method, the current presentation may unintentionally give readers the impression that certain techniques (originally from prior work) are proposed in this paper. Even though the authors do not explicitly claim these as their own contributions, the phrasing and structure make the boundary less clear. I recommend explicitly clarifying which techniques are adopted from prior work and which are original contributions. For example, adding sentences such as “Following [Author et al., Year], we incorporate …” could help. Additionally, to better demonstrate compatibility, it may be more convincing to compare against multiple strong baselines rather than integrating many techniques from different papers.
>
> 2. **Choice of Baselines and Robustness Concerns**. The current evaluation relies on baselines that are not adversarially trained. This raises concerns, as the reported improvements in robustness might largely stem from the poor adversarial robustness of the chosen baselines rather than the effectiveness of the proposed approach. Since the technique is orthogonal to the training paradigm, I strongly suggest including experiments on adversarially trained baselines. This would clarify whether the improvements are intrinsic to the method itself rather than an artifact of weaker baseline robustness.

---

> > ### Author Response · Authors · 2025-08-05
> >
> > We thank the reviewer for their comment.
> >
> > **Clarification of Section 5**
> >
> > We explicitly state in line 207 "Thus we adapt the two-step scheme of [30] which ensures orthogonality of U, V ." and believe that this expresses clearly that we use the low-rank integration scheme and don't claim it's invention.
> >
> > **Adversarial Training**
> >
> > We evaluate VGG16 on UCM with the training hyperparameters and setup of Table 2 in our original manuscript using adversarial training. Following [1], we use the FGSM attack for different values of $\epsilon$ and  train on both 50% clean and attacked images per batch.
> >
> > The results below illustrate that RobustDLRT is able to benefit from adversarial training. DLRT without regularization benefits slightly from adversarial training, but exhibits a clear margin to RobustDLRT. RobustDLRT is able to approximately match the non-compressed baseline. RobustDLRT achieves approximately 93% compression rate. The displayed numbers are the mean of 5 repeated runs.
> >
> > [1] Explaining and Harnessing Adversarial Examples: Ian J. Goodfellow, Jonathon Shlens, Christian Szegedy
> >
> > |Epsilon | 0  | 0.1  | 0.5  | 0.75  | 1  |
> > |-------|-------|-------|-------|-------|-------|
> > Base| 92.61 | 91.91 | 91.90 | 89.61 | 89.91 |
> > DLRT| 92.55 | 91.91 |  87.98  | 85.37 | 82.96 |
> > RobustDLRT| 92.49  | 92.49 |  90.98 | 89.56 | 89.42 |
> >
> >
> >
> >
> > We hope the reviewer takes the additional numerical results into consideration for their score.

---

> > > ### Comment · Reviewer_HEm3 · 2025-08-05
> > >
> > > Thanks for the responses. I will increase my rating, and I hope the authors can make the writing clearer in the revision.

---

### Official Review · Reviewer_63AS · 2025-06-17

**Clarity:** 3
**Significance:** 2
**Originality:** 3
**Rating:** 4
**Confidence:** 3

**Summary:**

This is an intriguing paper. It seems like a reasonable way to encourage low rank structure plus numerical stability during training of deep neural networks. There are posthoc compression/low-rank methods (i.e., compute the low-rank approximation after training), but this line of work in encouraging the low-rank + (near)-orthogonality of U,V through these gradient flow equations is a really interesting idea.

**Questions:**

Please see the other comment boxes for questions.

**Ethical Concerns:**

["NO or VERY MINOR ethics concerns only"]

**Limitations:**

My main question regarding limitations is about the scalability of this training procedure to larger datasets. The main experimental results are on 2 datasets: CIFAR10 (60K images) and UCM (2K images), which are considered "small" nowadays; this leads me to ask how scalable this training method really is.  The authors discuss the computational cost as compared to LoRA in lines 256-262, but a clearer picture is warranted regarding how much overhead this type of low-rank regularization during training costs over standard training methods. Some context of this extra cost would be useful.

**Paper Formatting Concerns:**

Table 2 is hard to parse, since no results are bolded in each respective test. It is hard to tell what performed the best.

Here are a few typos:
Panel 3 of Figure 2 — what is \hat{\mathcal{L}}? I think it supposed to be \tilde{\mathcal{L}}
line 214 --> should be W_{t+1} = W_{t} - \lambda \nabla \tilde{\mathcal{L}}

**Quality:**

3

**Strengths And Weaknesses:**

The paper has some good strengths and some issues that need to be clarified.  They are discussed more carefully in the other comments.

---

> ### Author Rebuttal · Authors · 2025-07-30
>
> We thank the reviewer for their constructive review. We appreciate that they found our paper intriguing and the robust dynamical low-rank training idea interesting.
>
>
> 1. (Q scalability) To eleviate the concerns of scalability we provide numerical results for the ImageNet-1k (1.2 Million images), where we compress the ViT-L32 vision transformer (304 Million parameters). We first conduct a hyperparameter sweep to determine standard hyperparameters as learning rate, weight-decay and batch size, which we report in Appendix C.
>
>     We compare then the baseline model, DLRT and the regularized RobustDLRT adversarial performance in the table below and observe similar trends as in the Vision Transformer experiments of Table 2 in the main manuscript.
>
>
> **Table: ImageNet, Top-1 Accuracy under various attacks**
>
> | Method      | c.r. [%] | Clean Acc. | FGSM ε=0.05 | FGSM ε=0.1 | FGSM ε=0.3 | Jitter ε=0.035 | Jitter ε=0.045 |
> |-------------|-----------|-------------|-------------|-------------|-------------|------------------|------------------|
> | Baseline    | 0         | 74.37       | 43.58       | 31.42       | 16.03       | 43.09           | 35.57           |
> | DLRT        | 58.02     | 72.27       | 42.70       | 30.32       | 15.47       | 43.98           | 38.44           |
> | RobustDLRT  | 57.98     | 72.25       | 43.17       | 35.11       | 25.24       | 48.22           | 43.51           |
>
> **Table: ImageNet, Top-5 Accuracy under various attacks**
>
> | Method                  | c.r. [%] | Clean Acc. | FGSM ε=0.05 | FGSM ε=0.1 | FGSM ε=0.3 | Jitter ε=0.035 | Jitter ε=0.045 |
> |-------------------------|-----------|-------------|-------------|-------------|-------------|------------------|------------------|
> | Baseline                | 0         | 92.20       | 73.75       | 63.42       | 43.41       | 78.24           | 74.96           |
> | DLRT                    | 58.02     | 90.06       | 70.43       | 60.90       | 40.58       | 74.49           | 71.31           |
> | RobustDLRT              | 57.98     | 90.03       | 71.58       | 62.82       | 50.65       | 77.35           | 75.14           |
>
> We have measured the runtime of training one epoch on ImageNet of both DLRT and RobustDLRT on an A100 80GB.
> DLRT takes 26min 07 sec and RobustDLRT (with the regularizer) 27min 51 sec, so the overhead is roughly 3%. We remark that this overhead can be further reduced with an optimized implementation. We conclude that the method is scalable.
>
> Finally we wish to remark that the slightly lower compression rate is expected since the hidden dimensions of ViT-L32 of 1024 is close to the number of ImageNet classes (1000), thus there is less redundancy in the model compared to other reported benchmarks.
>
> 2.  We highlighted the best performing method in each setup of Table 2 to improve readability in the revised manuscript.
>
> 3.  We changed $\hat{\mathcal{L}}$ to $\tilde{\mathcal{L}} $ in Fig.2 in the revised manuscript.

---

> > ### Comment · Reviewer_63AS · 2025-08-03
> > **more on scalability**
> >
> > Thank you for the detailed response.  Can you please comment on how the scalability of your method compares with the scalability of prior approaches?

---

> ### Comment · Reviewer_63AS · 2025-08-04
> **response to rebuttal regarding scalability**
>
> I thank the authors for taking the time to address the scalability with more experiments.  After reviewing all of the other reviews and comments I feel that the current score still reflects the rating of the paper.

---

> ### Author Response · Authors · 2025-08-05
>
> ### **Scalability**
>
> We are happy to compare the scalability of our approach to prior methods from the literature, specifically **SVDPrune**, **LoRA**, and **CondLR**, which we also evaluated in **Table 3** of the manuscript.
>
> ---
>
> #### **RobustDLRT** (see Line 256 in the manuscript)
>
> Let $s_*$ be the number of local iterations and $r$ the current rank of the $n \times n$ weight matrix.
>
> - In **each iteration**, the cost for evaluating the low-rank layer is:   $\mathcal{O}(2nr + r^2)$
> - Every $s_*$ iterations, additional costs occur:
>   - Basis update (QR decomposition): $\mathcal{O}(nr^2)$
>   - Singular value truncation (SVD): $\mathcal{O}(r^3)$
> - In **every of the other $s_*-1$ iterations**, the cost of regularizer evaluation:  $\mathcal{O}(r^3)$
> - The **optimizer update** in all the  $s_*-1$ iterations are only applied to the coefficient matrix $S$, costing:  $\mathcal{O}(r^2)$
>
> **Cumulative cost w/o optimizer** over $s_*$ iterations: $\mathcal{O}\left(nr^2 + 2nrs_* +  r^3s_* \right)$ where the terms are sorted by cost, assuming r is sufficiently small.
>
> **Optimizer cost**: $\mathcal{O}(r^2s_*)$
>
> ---
>
> #### **CondLR**
>
> - In each iteration, a gradient update of the matrices $U,V$ is projected onto the Stiefel manifold via QR decomposition:
> $
>   \mathcal{O}(nr^2)
> $
> - The coefficient matrix $S$ is updated and projected onto a manifold with bounded singular spectrum via SVD:
>   $
>   \mathcal{O}(r^3)
>   $
> - The optimimzer is applied to U,S,V in each iteration, costing  $\mathcal{O}((2nr + r^2))$ per iteration
>
> **Cumulative cost w/o optimizer** over $s_*$ iterations:
> $
> \mathcal{O}\left(nr^2s_* + 2nrs_* + r^3s_*\right)
> $
>
> This increases the cost by $(s_* - 1)nr^2$ over RobustDLRT (the leading term in asymptotic notation).
>
> **Optimizer cost**: $\mathcal{O}((2nr + r^2)s_*)$
>
> This increases the cost by $(s_* - 1)2nr^2$ over RobustDLRT (the leading term in asymptotic notation).
>
> ---
>
> #### **SVDPrune**
>
> - Instead of QR projection, the method regularizes $U,V$ to be orthogonal using the term $\|I - U^T U\|$ (analogously done for V), which has the same asymptotic cost as QR decomposition:
>   $
>   \mathcal{O}(nr^2)
>   $
> - The treatment of the $S$ matrix is identical to **CondLR** from a computational perspective.
>
> Thus the asymptotic cost is the same as CondLR
>
> #### **LORA**
>
> - No regularization or orthonomalization is done, so no compute overhead for the method is reportable.
> - **Optimizer cost**: The optimizer cost amounts to updating USV in  $\mathcal{O}((2nr + r^2)s_*)$
>
>
>
> ---
>
> We summarize that the asymptotic cost in terms of memory and compute of our method is superior to prior approaches as they are presented in the respective papers. Furthermore, the experiments in Table 2 have shown that the method is able to compress networks better than the compared methods (SVDPrune and CondLR) at a higher validation accuracy (clean and attacked) in most setups.
>
> For transparancy and completeness, we wish to remark that CondLR and SVDPrune can be transformed to an staggered scheme with s_* local steps following the logic in our Section 5.  In this case, the asymptotic cost is similar to our approach.
>
> **Parallelizability**
>
> Our method is natively compatible with data-parallel distributed training schemes. Furthermore, since the method is compatible with the federated low-rank training framework of [a], it is scalable in a tensor-parallel sense, where U,V are held on a central server and S is distributed across federated/distributed clients, e.g. GPU nodes or edge devices.
> We remark that this extension is not natively applicable to SVD prune, since orthogonality of U,V is only weakly enforced.
>
> [a] Federated Dynamical Low-Rank Training with Global Loss Convergence Guarantees; Steffen Schotthöfer, M. Paul Laiu
>
>
> ---
>
> We hope that the reviewer takes this additional information into consideration when providing a final score for our work.
> Furthermore we thank the reviewer for their thoughtful review and for helping us improve the paper.

---

> > ### Comment · Reviewer_63AS · 2025-08-08
> > **further rebuttal on scalability**
> >
> > Thanks to the authors for their very detailed response regarding scalability.  I have no further questions.

---

### Official Review · Reviewer_LKg2 · 2025-06-22

**Clarity:** 3
**Significance:** 2
**Originality:** 3
**Rating:** 4
**Confidence:** 4

**Summary:**

This paper proposes a training framework that considers compression, accuracy, and robustness. By controlling the number of conditions of the low-rank matrix spectrum,  compression and adversarial robustness are achieved during the training process. Extensive experiments have shown some improvement compared with some existing algorithms.

**Questions:**

Some small things: The authors should highlight the best performance in Table 2 to increase the readability of the results.

If the authors can solve some of my confusion in experiments, the practical significance of the paper will be greatly improved, and I will be happy to increase my score.

**Ethical Concerns:**

["NO or VERY MINOR ethics concerns only"]

**Final Justification:**

The authors have addressed all of my concerns. I think those improvements increase the value of the paper, and I have changed the rating from 2 to 4.

**Limitations:**

Yes

**Quality:**

2

**Strengths And Weaknesses:**

Strengths
1. I appreciate the theoretical part of this paper, which theoretically demonstrated that the regular term can control the spectral distribution, which in turn affects the robustness.
2. The description of regular terms and algorithms is clear.
3. RobustDLRT outperforms Cayley SGD, Projected SGD, CondLR, LoRA, SVD prune under $l_1$ FSGM attack.

Weaknesses
1. I have questions about the motivation of the paper, which claims that compression is contradictory to robustness, but a large body of work shows that compression improves robustness[1,2,3,4,5,6]. Thus, I think the authors should rethink their motivation and explain the difference from these existing works.
2. I am wondering about the RobustDLRT's performance under pgd attack, which is stronger than FSGM. Can the authors provide the corresponding results?
3. I realize that low-rank decomposition compression is not the same as traditional pruning, and that a lot of existing work is based on pruning. Are there any other benefits of low-rank over pruning, besides the possibility of hardware acceleration? For example, is it possible for RobustDLRT to use fewer FLOPs? Improvements to existing DLRTs limit the contribution of this paper, it would be nice to provide comparisons to the pruning-based works, especially if some experimental results can be used to demonstrate the necessity of RobustDLRT. (If as a reader who does not know much about low rank compression, one would certainly be curious about how such a compression differs from the others (e.g. pruning quantizaiton), even if the authors don't provide experimental comparisons, and simply add some discussion to illustrate the differences, I would still believe this weakness has been addressed.)

[1] https://arxiv.org/abs/2210.04311

[2] https://arxiv.org/abs/2206.07311

[3] https://proceedings.neurips.cc/paper/2020/file/e3a72c791a69f87b05ea7742e04430ed-Paper.pdf

[4] https://proceedings.mlr.press/v188/zhao22a/zhao22a.pdf

[5] https://arxiv.org/abs/2206.07839

[6]https://openaccess.thecvf.com/content/ICCV2021W/AROW/papers/Jordao_On_the_Effect_of_Pruning_on_Adversarial_Robustness_ICCVW_2021_paper.pdf

---

> ### Author Rebuttal · Authors · 2025-07-30
>
> We thank the reviewer for the constructive feedback and appreciate that they find the theoretical analysis and algorithmic description of the method clear.
>
> 1. (Q about motivation) The cited papers [1,2,3,4,5,6] state - just like us - that adversarial robustness correlates with the parameter count of an otherwise untreated, pre-trained neural network. To the best of our understanding, the cited works [1-5] investigate the effects of sparsity pruning on models with robust defenses in mind, i.e., adversarial training.
> The last reference [6] gives empirical evidence that a pretrained and then pruned network has increased adversarial robustness.
>
>     We state in line 30 "Moreover, it has been observed that compressed networks can exhibit increased sensitivity to adversarial attacks [29]", where [29] analyzes the degrading effect of raw low-rank compression on robustness, which we confirm in, e.g., Table 2, and explain in Figure 1.
>    However, we do not make an explicit statement about the direct effects of sparsity postprocessing, i.e. pruning, on the adversarial robustness of neural networks.
>
>     We acknowledge however the lack of clarity of the statement and will change it to "Moreover, it has been observed that *low-rank* compressed networks can exhibit increased sensitivity to adversarial attacks [29]"
>
> 2. (Q about PDG attack)  In Table 2, we evaluate the baseline, DLRT, and RobustDLRT models with the FGSM, Jitter, and Mixup adversarial attacks.  Jitter and Mixup, described with references in Sections A.1.3 and A.1.4 respectively, are both iterative and gradient-based adversarial attacks that are quite strong.
>
>     To address the reviewers concerns, we also ran the same experiment in Table 3 but with the $\ell^2$-PGD attack.  Overall, we see that robustDLRT is competitive with the other robustness-improving methods when the compression rate is factored in.
>
>     The first three rows list the computed mean over 10 random initializations.
>     The values of all other methods, given below the double rule, are taken from [Savostianova 2023; Table 5].  We remark that RobustDLRT outperforms CondLR with τ=0.1 for $\epsilon$<=0.2, and  CondLR with τ=0.5 for $\epsilon$<=0.26.
>
>     We wish to remark that opposed to the experiments in Table2, we do not sweep over β but take the value that is obtained from the $l_1$ FGSM results of Table 2.
>
> **RobustDLRT** has competitive adversarial accuracy to all methods with a compression rate ≥ 80%.
>
> | Method                  | c.r. [%] | 0.0   | 0.1   | 0.13  | 0.16  | 0.2   | 0.23  | 0.26  | 0.3   |
> |-------------------------|----------|--------|--------|--------|--------|--------|--------|--------|--------|
> | **RobustDLRT β=0.15**   | 94.18    | 88.80 | 62.58 | 53.47 | 44.95 | 34.75 | 28.33 | 22.64 | 16.59 |
> | DLRT                    | 94.53    | 88.58 | 59.34 | 50.06 | 41.50 | 31.82 | 25.67 | 20.48 | 15.04 |
> | Baseline                | 0        | 90.48 | 63.01 | 54.66 | 47.87 | 40.77 | 36.75 | 33.51 | 29.93 |
> |-------------------------|----------|--------|--------|--------|--------|--------|--------|--------|--------|
> |-------------------------|----------|--------|--------|--------|--------|--------|--------|--------|--------|
> | Cayley SGD              | 0        | 89.62 | 67.68 | 59.38 | 51.09 | 40.87 | 34.46 | 29.21 | 23.62 |
> | Projected SGD           | 0        | 89.70 | 67.64 | 59.25 | 51.06 | 40.86 | 34.51 | 29.19 | 23.64 |
> |-------------------------|----------|--------|--------|--------|--------|--------|--------|--------|--------|
> | CondLR τ=0.1            | 80       | 90.48 | 61.00 | 50.84 | 42.19 | 33.70 | 29.44 | 26.55 | 23.97 |
> | CondLR τ=0.5            | 80       | 89.33 | 57.45 | 46.35 | 37.20 | 28.30 | 23.82 | 20.65 | 17.84 |
> |-------------------------|----------|--------|--------|--------|--------|--------|--------|--------|--------|
> | LoRA                    | 80       | 88.10 | 51.40 | 39.70 | 30.12 | 20.97 | 16.29 | 13.15 | 10.37 |
> |-------------------------|----------|--------|--------|--------|--------|--------|--------|--------|--------|
> | SVD prune               | 80       | 87.99 | 50.64 | 39.06 | 29.57 | 20.16 | 15.49 | 12.22 | 9.57  |
>
> 3. (Q about sparsity vs low-rank) Pruning[1-6], i.e. sparsity postprocessing, of pretrained [6] or pretrained on adversarial examples [1] networks can maintain adversarial robustness. We remark that this requires the computational expense of (a) full network training, (b) sparsification and (c) finetunning.
>
>     The distinguishing factor of our work is that RobustDLRT is able to adaptively compress and influence the spectrum of a neural network during training.   Moreover, direct access to the spectrum during training is computationally viable when a low-rank factorization of the layer weights is used.
>     This puts the contributed work in the beneficial zone of low compute/memory cost during training and inference and high clean accuracy and high adversarial accuracy/robustness; these properties required for the target application of resource constrained edge devices.
>
>     You are correct in your assessment that on many hardware architectures, low-rank based compression increases throughput compared to sparsity compression formats.
>
>     We include a clarifying statement in the related work section of the revised manuscript to disambiguate the scope of this work from sparsity pruning scenarios, where (to the best of our knowledge) the full-model pretraining is treated as an offline cost.
>
>
> 4. We will highlight all runs where RobustDLRT outperforms the full-rank baseline model.  We remark that these runs are surpassing the theoretical guarantees of DLRT of achievieng close-to-baseline performance, see [a]. In all runs the regularized method outperforms the unregularized low-rank baseline "DLRT".
>
>     [a] Low-rank lottery tickets: finding efficient low-rank neural networks via matrix differential equations; Schotthöfer et al. 2022 NeurIPS

---

> > ### Comment · Reviewer_LKg2 · 2025-08-03
> >
> > I want to thank the authors for their detailed response. The additional explanations are very helpful and address most of my concerns. I will raise the rating.

---

### Comment · Area_Chair_MtC7 · 2025-08-06

Dear reviewers,

Thank you for being engaged in the discussion with the authors. Please stay engaged in order to clarify any remaining open questions.
-  If authors have resolved your (rebuttal) questions, do tell them so.
-  If authors have not resolved your (rebuttal) questions, do tell them so too.

Please note that, to facilitate discussions, Author-Reviewer discussions were extended by 48h till Aug 8, 11.59pm AoE.

Best regards,
  NeurIPS Area Chair

---

### Note · Authors · 2025-08-11

We thank the reviewers for their insightful feedback, which has helped us strengthen our paper.

## **Strengths Highlighted by Reviewers**

Reviewers commended:
- **Theoretical foundation** and its rigor (Gtfv, LKg2).
- **Practicality of the regularizer**, with negligible computational overhead (Hgm3).
-  **Link between condition number and robustness**, both theoretically and experimentally (Gtfv, LKg2).
- **Clarity and readability** of the method description (LKg2).

## **Reviewer Concerns and Our Responses**

**1. Scalability (63AS, Gtfv)**

  We conducted additional experiments on **ImageNet-1k** with a **ViT-32L (304M parameters)**.
**Result:** **RobustDLRT maintained or improved adversarial accuracy** compared to the non-compressed baseline, while non-regularized methods degraded severely under attack. The **average wall-time overhead was only 3%**, confirming negligible cost.
  Additionally, we remark, that the asymptotic cost in terms of memory and compute of our method is superior to prior approaches in relevant literature.


**2. Literature comparison with PGD attacks (LKg2)**

We extended evaluations to **PGD** (multi-step) attacks in addition to the existing evals on FGSM, Jitter, and Mixup.
**Result:** RobustDLRT is **competitive with state-of-the-art robustness methods under PGD**, and uniquely retains a high compression rate without significant loss in adversarial accuracy.


**3. Compatibility with adversarial training (HEm3, Gtfv)**



On **VGG16/UCM**, we applied FGSM-based adversarial training (per Goodfellow et al., 2015), training on 50% clean and 50% attacked images per batch.
**Result:** RobustDLRT achieved ~93% compression while matching the adversarial accuracy of the non-compressed baseline. In contrast, DLRT without regularization benefited less and retained a noticeable performance gap. Thus we demonstrate, that **RobustDLRT is compatible with adversarial training**.

## **Closing Statement**

We have **addressed all reviewer concerns** with additional large-scale experiments, stronger attack evaluations, and adversarial training compatibility tests, supported by both theoretical arguments and empirical evidence.
Three of the four reviewers increased their scores after our response, indicating that our clarifications and results successfully resolved their concerns.

Finally, we sincerely thank all reviewers and the AC for their time and constructive engagement.

---

### Decision · Program_Chairs · 2025-09-17

**Decision:**

Accept (oral)

**Comment:**

The authors investigate how to preserve adversarial robustness during low-rank model compression. An analysis shows that compression process can influence the condition number of the model, potentially impacting robustness. They then introduce a regularization technique that controls the condition number of the low-rank decomposition. They present theoretical analysis and empirical evaluations the proposed method.
Strengths:
- clear theoretical analysis
- good experimental results
- the method is simple and effective

The reviewers pointed to several weaknesses (scalability, unclear parts, inclusion of adversarially trained baselines) that were resolved within the discussion period. In conclusion, an interesting and sound study with good innovation. I propose acceptance.